# Methods and considerations for estimating parameters in biophysically detailed neural models with simulation based inference

**Nicholas Tolley**[1]*, **Pedro L. C. Rodrigues**[2], **Alexandre Gramfort**[3]☉, **Stephanie R. Jones**[1]☉

**1** Department of Neuroscience, Brown University, Providence, Rhode Island, United States of America, **2** Université Grenoble Alpes, Inria, CNRS, Grenoble INP, LJK, Grenoble, France, **3** Université Paris-Saclay, Inria, CEA, Palaiseau, France

☉ These authors contributed equally to this work.
* nicholas_tolley@brown.edu

**Data Availability Statement:** All code for analysis and figure generation is available at https://github.com/ntolley/hnn_sbi_examples. Data produced

## Abstract

Biophysically detailed neural models are a powerful technique to study neural dynamics in health and disease with a growing number of established and openly available models. A major challenge in the use of such models is that parameter inference is an inherently difficult and unsolved problem. Identifying unique parameter distributions that can account for observed neural dynamics, and differences across experimental conditions, is essential to their meaningful use. Recently, simulation based inference (SBI) has been proposed as an approach to perform Bayesian inference to estimate parameters in detailed neural models. SBI overcomes the challenge of not having access to a likelihood function, which has severely limited inference methods in such models, by leveraging advances in deep learning to perform density estimation. While the substantial methodological advancements offered by SBI are promising, their use in large scale biophysically detailed models is challenging and methods for doing so have not been established, particularly when inferring parameters that can account for time series waveforms. We provide guidelines and considerations on how SBI can be applied to estimate time series waveforms in biophysically detailed neural models starting with a simplified example and extending to specific applications to common MEG/EEG waveforms using the the large scale neural modeling framework of the Human Neocortical Neurosolver. Specifically, we describe how to estimate and compare results from example oscillatory and event related potential simulations. We also describe how diagnostics can be used to assess the quality and uniqueness of the posterior estimates. The methods described provide a principled foundation to guide future applications of SBI in a wide variety of applications that use detailed models to study neural dynamics.

## Author summary

A central problem in computational neural modeling is estimating model parameters that can account for observed activity patterns. While several techniques exist to perform parameter inference in special classes of abstract neural models, there are comparatively

from simulation generating code, and experimentally recorded MEG Beta Event data, is available on the Open Science Framework at is available on the Open Science Framework at https://doi.org/10.17605/OSF.IO/VZ97X Results of diagnostic analyses of the RC circuit and simplified HNN examples for all parameters can be found in the files hnn_rc_diagnostics.csv and rc_circuit_diagnostics.csv in the same repository.

**Funding:** This work was supported by the Chateaubriand Fellowship (https://chateaubriand-fellowship.org), computing resources offered by the Neuroscience Gateway (NSG; https://www.nsgportal.org), l'Agence nationale de la recherche (ANR; https://anr.fr; grant number ANR-20-CHIA-0016 awarded to A.G.), the National Institute of Health (NIH; https://www.nih.gov; grant numbers R01AG076227 and RF1MH130415 awarded to S.R.J., and grant number U24EB029005 awarded to the NSG team), and the National Science Foundation (NSF; https://www.nsf.gov; grant numbers 1935749 and 1935771 awarded to the NSG team). S.R.J. received salary support from NIH grant numbers R01AG076227, RF1MH130415, and U24EB029005. A.G. received salary support from ANR grant number ANR-20-CHIA-0016. The funders had no role in study design, data collection and analysis, decision to publish, or preparation of the manuscript.

**Competing interests:** The authors have declared that no competing interests exist.

few approaches for large scale biophysically detailed neural models. In this work, we describe challenges and solutions in applying a deep learning based statistical framework to estimate parameters in a biophysically detailed large scale neural model, and emphasize the particular difficulties in estimating parameters for time series data. Our example uses a multi-scale model designed to connect human MEG/EEG recordings to the underlying cell and circuit level generators. Our approach allows for crucial insight into how cell-level properties interact to produce measured neural activity, and provides guidelines for diagnosing the quality of the estimate and uniqueness of predictions for different MEG/EEG biomarkers.

## Introduction

Biophysically detailed neural modeling is a fundamental and established framework to study fast time scale neural dynamics [1, 2]. While challenging to construct, advances in computational resources have enabled the proliferation of detailed models from principled models of single neurons [3] to large scale biophysically detailed neural networks [4] that enable multi-scale interpretation from cell spiking to local field potentials to macroscale magneto- and electroencephalographic (MEG/EEG) signals [4–7]. Numerous detailed models are now openly distributed to encourage their use and expansion [4, 5, 7–13]. A common goal of detailed neural modeling is to infer biophysical parameters in individual cells and/or network connections that can account for observed changes in neural activity over time. Given the large-scale nature of any detailed model, parameter inference is an inherently challenging and unsolved problem. The difficulty of parameter inference is closely tied to the level of biophysical detail in the model, as the number of parameters increases with more realistic models. In practice, parameters can not all be estimated simultaneously, but rather model elements are estimated in a serial fashion (e.g. cell dynamics followed network connectivity) and fixed. Then, a limited set of parameters are chosen as the target for estimation. This limited set is chosen based on a prior hypothesis that a certain set of unknown parameters can be estimated based on data features. Even with this limited set, the parameter estimation process is complex. The problem is confounded by the fact that there may be many parameter configurations that produce an equally good representation of the data [14]. In this study, we focus on the latter problem of identifying parameter indeterminacies in a limited set of parameters, as to date there is no means to estimate large numbers of parameters at once. The goal is not to present a method to mitigate indeterminacies, but develop a framework that can fully represent them. Identifying unique biophysically constrained parameter sets that can account for observed neural dynamics, and differences across experimental conditions, is essential to the meaningful use of large scale biophysically detailed models for neuroscience research. For example, if you want to use a biophysically detailed model to infer circuit level mechanisms generating an EEG waveform that is a biomarker of a healthy compared to neuropathological condition, you need a way not only to estimate the parameter distributions that can generate the waveforms but also to assess if the distributions are distinguishable.

A powerful approach to estimate parameters in neural models is Bayesian inference. There is an extensive history of research applying Bayesian inference, and specifically the algorithm of variational inference, to estimate parameters in reduced models of neural activity, for example in the Dynamic Causal Modeling (DCM) [15] framework that relies on reduced "neural mass models". However, while compatible with reduced models that are mathematically tractable, the algorithm of variational inference is not compatible with detailed biophysical models

due to their computational complexity, and specifically lack of access to a known likelihood function (see Discussion). In recent years, simulation based inference (SBI) has been proposed as an alternative Bayesian inference framework to estimate parameters in detailed neural models. SBI overcomes the challenge of not having access to a likelihood function by leveraging advances in deep learning to perform density estimation [16–18]. An advantage of SBI is that it only relies on a dataset of simulations from the model being investigated, rather than requiring knowledge of a parameter likelihood function, which is typically not accessible in large-scale biophysically detailed models. From this dataset, a neural density estimator (i.e, artificial neural network specifically made to approximate probability distribution functions) is then trained to learn a mapping of observed neural dynamics (e.g. time series waveforms) to corresponding model parameter distributions. Another advantage is that SBI estimates a full distribution over model parameters that may account for the data and provides information about parameter interactions [14, 18]. This information is not possible with optimization techniques that have historically been used in large-scale biophysical models such as COBYLA [5, 19] and genetic algorithms [3, 20], which estimate only a single parameter configuration that best fits the data. Fig 1 outlines the overall SBI workflow to estimate parameters and possible parameter combinations that can reproduce recorded neural activity.

While the substantial methodological advancements offered by SBI are promising, and have been applied to estimate parameters in small biophysically detailed neuron models [18, 21] and in models with reduced representations of neural dynamics [22], there is currently little guidance on how these methods should be used with large-scale biophysical models, with the notable exception of [23] offering a thorough discussion of using SBI on simplified models coupled in a large-scale brain network. Guidance is particularly lacking in the context of performing inference on neural time series data, and in comparing estimates for data from different experimental conditions. In this paper, we provide guidelines on how SBI can be applied to estimate parameters underlying time series waveforms generated by biophysically detailed neural models. We emphasize the importance of the first steps of (i) identifying the parameters and ranges over which the inference process will be performed (i.e. prior distribution), which necessarily depends on user-defined hypotheses, and (ii) of selecting informed summary statistics of the waveform activity. We also describe how diagnostics can be used to assess the uniqueness and quality of the posterior estimates and to assess the overlap of distributions from estimation applied to two different waveforms. These evaluation steps are particularly important to resolve the uniqueness of distributions for two or more waveforms.

We begin with a simplified example of a non-linear resistor-capacitor circuit, and then extend the results to an example large-scale biophysically detailed modeling framework that was developed by our group to study the multi-scale neural origin of human MEG/EEG signals, namely the Human Neocortical Neurosolver (HNN) [5]. The foundation of HNN is a biophysically detailed model of a neocortical column, with layer specific synaptic activation representing thalamocortical and cortico-cortical drive (Fig 2). HNN has been applied to study the cell and circuit origin of commonly measured MEG/EEG signals, including low frequency oscillations (e.g. Beta Events [24] and event related potentials (ERPs) [25, 26]), along with differences across experimental conditions [25–29]. We demonstrate applications of SBI to estimate parameter distributions that can account for variation in example Beta Events based on empirical data and in simulated ERP waveforms with selected parameter priors (see Step 1 in Fig 1) based on our previous studies [24–26]. We show that due to the model complexity some parameters can be inferred uniquely while others are indeterminate. The methods described and proof of concept examples provide a principled foundation to guide future applications of SBI in a wide variety of applications that use detailed models to study neural dynamics.

## What model parameters and possbile parameter combinations can produce recorded neural activity?

**1) Identify relevant model parameters and parameter ranges**

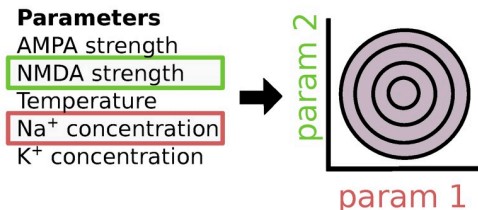

**2) Randomly sample from parameter ranges to simulate "real-world" neural waveforms**

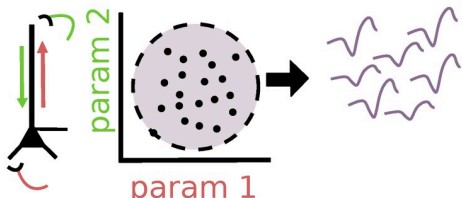

**3) Select summary statistics to describe the waveform**

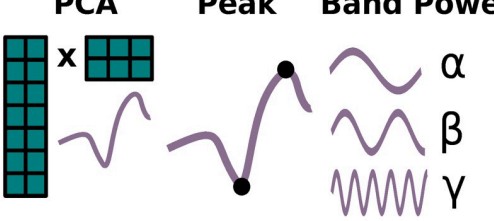

**4) Train artifical neural network that can indentify parameter distributions constrained by summary statistics**

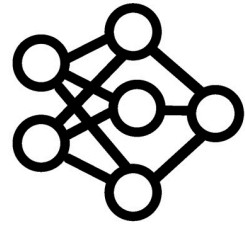

**5) Apply ANN to estimate parameter distributions for waveforms of interest**

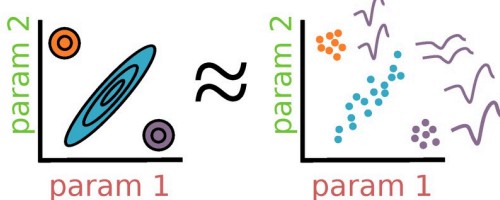

**6) Run diagnostics and compare parameter distributions**

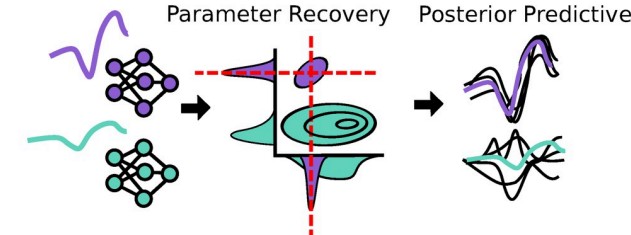

**Fig 1. Graphical abstract.** Summary of SBI workflow used to infer model parameters that can account for recorded neural dynamics. 1) A prior distribution of assumed relevant model parameters and ranges is constructed. 2) A dataset of simulated neural activity patterns is generated with parameters sampled from the prior distribution. 3) User defined summary statistics are chosen to describe waveform features of interest. 4) A specialized deep learning architecture is trained to learn the mapping from neural activity constrained by summary statistics to underlying model parameters. 5) Specific neural activity patterns of interest are fed into the trained neural network, which subsequently outputs a distribution over the potential underlying model parameters. 6) Parameter estimates for different waveforms can be compared through diagnostics like parameter recovery (if the ground truth is known), or posterior predictive checks.

## Materials and methods

Below we provide a summary of the primary techniques used in this work. Specific aspects are emphasized to provide better context on the significance/motivation of analyses performed. We invite readers to refer to the citations for a more thorough treatment of each subject. In particular, the software publication detailing HNN [5], and a study demonstrating the use of

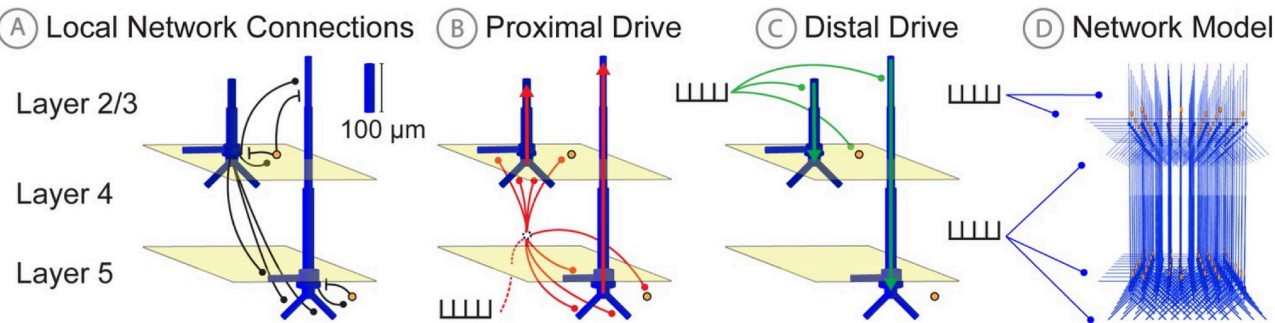

**Fig 2. HNN model schematic. A**: Local network connections of the HNN model include: 1) excitatory AMPA and NMDA synaptic connections (black circles) originating from pyramidal neurons (blue), and 2) inhibitory GABA$_A$ and GABA$_B$ synaptic connections (black lines) originating from inhibitory interneurons (yellow). **B**: Proximal exogenous input connection pattern. **C**: Distal exogenous input connection pattern, see text for further description. **D**: 3D rendering of full neocortical column model. Figure adapted from Neymotin et al. 2016 [5].

SBI on classical neural models [18]. We also detail an RC circuit example, which is a building block of HNN-type models, and which offers a SBI setup with time series inputs and the presence of indeterminacies.

### Resistor-capacitor circuit simulations

Resistor-capacitor (RC) circuit simulations were performed using the `odeint` ordinary differential equation (ODE) solver of the `SciPy` Python package. A more thorough description of how RC circuit simulations were performed can be found in the results section RC circuit: A simple example to describe indeterminacies that can occur with time series inference.

### Human Neocortical Neurosolver

Biophysical modeling of cortical activity underlying MEG/EEG signals was performed using the Human Neocortical Neurosolver (HNN) [5]. The standard HNN model used in this study and described in [26] is composed of 100 pyramidal neurons and 33 inhibitory neurons in both layers 2/3 (L2) and 5 (L5) for a total of 266 neurons and represents a localized small patch of neocortex (Fig 2). To accurately reproduce macroscale electrical signals, HNN utilizes multi-compartment pyramidal neuron models [30], and synchronous intracellular current flow of aligned L2 and L5 pyramidal neuron dendrites is assumed to be the generator of the primary electrical current sources underlying recorded extracranial MEG/EEG signals, due to their length and parallel alignment [31]. Inhibitory basket cells are modeled as single compartment point neurons given their negligible impact in producing the recorded electrical currents, but are none-the-less crucial to the local network dynamics (see [5] for further background). Pyramidal cells in the model connect to other cells with both AMPA and NMDA synapses, while basket cells produce GABA$_A$ and GABA$_B$ synaptic connections.

In addition to the local circuitry, HNN models extrinsic inputs to the column via layer specific synaptic excitatory drives generated by predefined patterns of action potentials presumed to come from exogenous brain areas (i.e., thalamus and other cortical regions). In general these are referred to as proximal and distal drives, reflecting the effective location of the synapses on the pyramidal neuron proximal and distal dendrites. The proximal drive reflects so called feedforward inputs from lemniscal thalamus, and the distal drive reflecting inputs from either non-lemniscal thalamus [32, 33], or "feedback" connections from other cortical regions. These proximal and distal drives induce excitatory post-synaptic currents that drive current

flow up and down the aligned pyramidal cell dendrites (see red and green arrows in Fig 2) that can generate positive and negative deflections in the primary electric current dipole of source localized MEG/EEG signals. Intracellular current flow due to these extrinsic inputs, as well as induced local spiking dynamics, combine to produce the recorded MEG/EEG signal.

The results detail 3 examples of problems suited for parameter inference in HNN. In the first two examples, the parameters to be estimated control either the timing or strength of extrinsic inputs. In the first example which demonstrates HNN simulations mimicking an RC circuit, the target of estimation is the parameter vector $\theta_0 \in \mathbb{R}^3$, where the first two parameters control the strength of a single spike of excitatory proximal and distal input (in units of nS for the synaptic conductance $\bar{g}$), and the third parameter $\Delta t$ controls the timing between the two inputs (in units of milliseconds (ms)). In the second example which demonstrates Beta Events, the target of estimation is $\theta_0 \in \mathbb{R}^2$ where the parameters control the variance (ms$^2$) of a Gaussian distribution from which 10 spikes are drawn for the proximal and distal inputs. In the final example which demonstrates event related potentials, the target of estimation is $\theta_0 \in \mathbb{R}^4$, where the parameters control the synaptic strength (nS) of excitatory/inhibitory local connections onto the proximal/distal dendrites of L5 pyramidal neurons.

## Posterior diagnostics

When working with a posterior approximation $\Phi$, it is useful to characterize its behavior in different regions of the parameter space. For a given parameter configuration $\theta_0$ and simulated output $x_0 \sim p(x \mid \theta_0)$, we quantify how concentrated $\Phi(\theta \mid x_0)$ is around $\theta_0$. We refer to this quantity as the parameter recovery error (PRE) and define it as the Wasserstein distance between the $k$-th marginal of $\Phi(\theta \mid x_0)$, and a Dirac delta centered at $\theta_0[k]$. This can be empirically estimated as per

$$\mathrm{PRE}_k(\Phi, \theta_0) = \frac{1}{N} \sum_{i=1}^{N} \left( \theta_i[k] - \theta_0[k] \right)^2 \tag{1}$$

where we generate $N$ samples $\{\theta_1, \ldots, \theta_N\} \sim \Phi(\theta \mid x_0)$ from our posterior approximation conditioned at $x_0 \sim p(x \mid \theta_0)$, which is an observation from the simulator at ground truth $\theta_0$. Note that $\theta_0$ is composed of $k$ distinct values for each individual parameter, therefore there are $k$ distinct PRE values. Additionally, each parameter $\theta_0[k]$ is mapped from its range defined in the prior distribution, to the range (0,1). Therefore, the maximum PRE is 1.0, indicating the worst possible recovery, whereas a PRE of 0.0 indicates perfect recovery.

The previous diagnostic quantified how well $\Phi$ represents the relationship between $x_0$ and $\theta_0$ in terms of the parameter space. Alternatively, we can assess the relationship in the observation space. Specifically, given samples from our approximate posterior $\theta_i \sim \Phi(\theta \mid x_0)$, we generate simulations $x_i \sim p(x \mid \theta_i)$, and assess how close $x_i$ is to the conditioning observation $x_0$. This is known as a posterior predictive check (PPC) [34, 35] and is quantified in this manuscript as the root mean squared error between $x_0$ and the $x_i$ as per

$$\mathrm{PPC}(x_i, x_0) = \sqrt{\frac{1}{N} \sum_{i=1}^{N} \|x_i - x_0\|^2} \ . \tag{2}$$

We note that this is a simplified treatment of how to perform a PPC, as observation and neural noise may significantly change the interpretation of model fit. A more robust assessment of PPC in the context of systems neuroscience modeling can be found in [36] which describes measures such as the Watanabe–Akaike information criterion (WAIC), as well as alternative measures to PRE for assessing posterior fit such as posterior shrinkages.

Lastly, in many applications it is useful to characterize how well two distributions can be distinguished from one another. To this end we introduce the distribution overlap coefficient (OVL) [37] which varies on a scale of [0, 1] such that 1 indicates complete overlap, and 0 indicates no overlap. Given our approximate posterior $\Phi$, we define OVL as:

$$\text{OVL}_k(\Phi, x_0, x_1) = \int \min\Big(\Phi(\theta[k] \mid x_0), \Phi(\theta[k] \mid x_1)\Big) d\theta \qquad (3)$$

where $x_0$ and $x_1$ are two different observations whose posterior distributions we seek to compare on the marginal distribution for the $k$-th parameter. To calculate the $\text{OVL}_k$ numerically we used an evenly spaced grid of $50^d$ for a prior distribution over $d \in \mathbb{N}_+$ parameters. $\text{OVL}_k$ operates similarly to Kullback–Leibler divergence and related measures, with the primary advantage being that it is bounded on the interval [0, 1] readily permitting identification of distributions with large and small amounts of distribution overlap.

## MEG Beta Events

Beta Event examples represent empirical human MEG data source localized to SI taken from the previous studies [24, 26, 38] and preprocessed as described in [24]. Subjects 7 and 5 from [24] were chosen as exemplars for large (blue) and small (orange) Beta Events, and here are referred to as subjects 1 and 2, respectively. The Beta Events used for inference were averaged over all recorded events for each subject that were included in the dataset: 378 events for subject 1, and 376 events for subject 2. The full dataset of source localized MEG Beta Events used in this study can be found in the Open Science Framework repository associated with this manuscript: https://doi.org/10.17605/OSF.IO/VZ97X.

## Prior distribution setup and sampling

Prior samples $\theta_i \in \mathbb{R}^d$ were generated by using the PyTorch Uniform distribution on the interval [0,1]. The values in each dimension were then linearly mapped to the range of their corresponding parameter values. For parameters specifying the maximum conductance $\bar{g}$ (nanosiemens, nS) of synaptic connections in HNN, the values were exponentiated in base 10 after being mapped to the appropriate range. This is to account for the saturating impact of conductance on model outputs, and is employed in previous SBI work with biophysical models [18].

## Simulation and SBI training

Prior samples and simulations were all generated and stored in the form of `NumPy` binary arrays before neural density estimator training. The `SBI` Python package was used for all neural density estimator training and posterior evaluation. A masked autoregressive flow architecture was utilized for approximation of the posterior distribution. Posteriors for all examples were trained using a dataset of 100,000 samples from the prior distribution. Gaussian white noise was added to training observations $x_i$ [39]. The variance of the Gaussian noise added to observations was 0.01 for RC circuit simulations, and 1e-5 for HNN simulations.

All analysis was performed on the Expanse supercomputing cluster managed by XSEDE and the Neuroscience Gateway. HNN simulations were generated using the Dask distributed scheduler configured for the SLURM workload manager. When distributed across 256 CPU cores, the simulated dataset for each example generally took <8 hours to generate. The neural density estimator took <20 minutes to converge when trained on the CPU of a single computing cluster node with 32 cores.

Diagnostic heatmaps were constructed by defining a grid over the support of the prior with a range of $[0.05, 0.95]^d$, with a resolution of 10 samples in each dimension $d$.

## Results

### Approach to applying SBI in biophysically detailed models that simulate time series data

Recently SBI has been established as an approach to estimate parameters in detailed biophysical models [18] that simulate time series data. This approach overcomes the challenges of applying Bayesian inference in highly detailed non-linear models, namely estimation of complex posterior distributions that exhibit parameter interactions, by leveraging recent advances in likelihood-free inference and deep learning [17]. We begin by reviewing the SBI process and providing the mathematical description of each of the steps outlined in Fig 1.

The primary goal of SBI in the context of our manuscript is to estimate parameters and possible parameter distributions that can account for an observed neural dynamic (e.g. time series waveform). In mathematical terms, this goal is stated as follows. Given an observation $x$ and model with parameters $\theta$, SBI seeks to create an approximation $\Phi(\theta \mid x)$ of a posterior parameter distribution $p(\theta \mid x)$ such that

$$\Phi(\theta \mid x) \approx p(\theta \mid x) \propto p(x \mid \theta) \; p(\theta) \; . \tag{4}$$

Bayes' rule specifies a closed form for the desired posterior distribution $p(\theta \mid x)$ as being proportional to the likelihood $p(x \mid \theta)$ multiplied with the prior $p(\theta)$ [40]. Unfortunately, in detailed biophysical models the likelihood function $p(x \mid \theta)$, which encodes the relationship between model parameters $\theta$ and outputs $x$, is often analytically intractable but can be approximated from a large number of simulations. The novelty of SBI is that it circumvents likelihood evaluations altogether, and instead approximates the posterior distribution directly from a simulated dataset of model outputs $x_i \sim p(x \mid \theta_i)$ with parameter values sampled from a user defined prior distribution $\theta_i \sim p(\theta)$. There are numerous approaches in the literature to achieve this goal, each with their own unique considerations, benefits, and challenges [41]. In this study, we use a deep learning architecture known as a conditional neural density estimator ($\Phi$), which is a function that takes an observation $x$ as input, and returns a probability density function defined over the parameter space. More specifically, the conditional neural density estimator utilizes normalizing flows following standard practices described in [17, 18]. The detailed steps in applying SBI (Fig 1) are as follows.

**Step 1: Define prior distribution.** SBI begins with the user choosing the parameters of interest to be estimated, and the range and statistical distribution of values (e.g. uniform distribution) those parameters can take. This constitutes the prior distribution $p(\theta)$ over model parameters $\theta$ to be inferred. The importance of a well-constructed prior cannot be understated, as it encodes the assumptions and hypotheses of the inference problem considered, and strongly impacts any resulting predictions. Creating a good prior distribution requires domain expertise to choose meaningful parameters, and a biologically realistic range of values. That is not to say that the predictions are predetermined by the prior, as uncertainty can be encoded using flat/uninformative priors where the probability mass is evenly spread over the desired parameter range. Nevertheless this aspect is highly important for detailed neural models where the inferred parameters represent a small subset of the total set of parameters.

**Step 2: Generate training data.** With the prior constructed, a simulated dataset of observations $x_{1:N}$ is generated by simulating a large number of $N$ time series using model parameters $\theta_{1:N}$ drawn from the prior parameter distribution.

**Steps 3-4: Training neural network based on chosen summary statistics to describe the time series waveform.** With the simulated dataset, a specialized deep learning architecture known as a conditional neural density estimator $\Phi$ is trained to approximate the posterior distribution $p(\theta|x)$ that can account for chosen summary statistics for any observation $x$. The output is a distribution of parameters which can generate simulations close to the summary statistics of the conditioning observation $x_0$. The neural density estimator $\Phi$ can be trained directly from the entire time series $x$, or from summary statistics $s = S(x)$ which constitute a lower dimensional vector of values. The choice of $s$ should aim to obtain $\Phi(\theta \mid s) \approx p(\theta \mid s) \approx p(\theta \mid x)$, meaning that the posterior estimates are well enough approximated from $s$ alone. A more in depth description of the choice and role of summary statistics is given after describing Steps 5-6.

**Steps 5-6: Estimate and compare parameter distributions for distinct waveforms.** With a trained neural density estimator, users can finally feed in new waveforms and assess the predicted parameter distributions underlying their generation. Diagnostics that assess the quality of the distribution can then be performed (see Materials and methods for details on the calculation of each diagnostic). If the ground truth parameters underlying the waveform of interest are known, users can calculate the dispersion of the posterior around this ground truth using parameter recovery error (PRE). This is known as face validity when using simulated data to check if the ground truth parameters can be recovered [42, 43]. If the ground truth is unknown, users can use posterior predictive checks (PPC) to assess if the parameter distribution consistently produces waveforms close to the conditioning observation. Finally, uniqueness of posterior estimates for two different waveforms can be assessed by directly comparing the overlap coefficient of the distributions (OVL).

**The important role of summary statistics in parameter inference.** Using the inference framework described above, in this study we emphasize the role of summary statistics and how their selection directly impacts predicted parameter distributions. Inference on models with full time series outputs is challenging because the observations are high dimensional. This challenge comes in the form of interpretability and model misspecification due the simulator not capturing finer characteristics of the real data generating process [44].

Summary statistics can either be hand-crafted, leveraging domain expertise and hypotheses regarding the data, or they can be automatically extracted. A summary statistic $s = S(x)$ is sufficient if all the relevant information for mapping the full observation to its underlying parameters is retained. More specifically, sufficiency is satisfied if $p(\theta \mid s) = p(\theta \mid x)$ [22, 41]. In practice, truly sufficient summary statistics are rare, but they can still provide a close approximation to inferences achieved when using the full observations $x$. A major advantage of hand-crafted features is that they can be readily interpretable, and come associated with hypotheses on their physiologic significance depending on previous research. Alternatively, full time series informed approaches like principal component analysis (PCA) may do a better job at retaining more complex relationships between observations and parameters. We note that as an alternative to PCA, automatic extraction of summary statistics through embedding networks are becoming increasingly common [17, 22, 45, 46]. However, a systematic analysis of the numerous architectures employed in this domain is outside the scope of this study.

Given the current ambiguity around summary statistic selection, principled approaches to compare alternate approximations of the posterior distribution are essential. To this end, we have constructed educational examples which allow for an intuitive understanding of the role of different summary statistics. Additionally, we introduce diagnostic analyses that can be used to quantitatively compare desirable properties of posterior approximations produced using different summary statistics, such as PPC, PRE, and OVL. Note that we use the term "desirable properties" as the goals of inference may differ depending on the use-case. For example,

predicting precise values for biophysical values (parameter recovery) may be one goal. Alternatively, characterizing the range of parameters consistent with a known biomarker may be another goal. These analyses are detailed in the examples below.

## RC circuit: A simple example to describe indeterminacies that can occur with time series inference

One of the most challenging aspects of likelihood free inference is that the models studied typically do not permit access to a ground truth posterior distribution, over which we can validate our inferences. To highlight this challenge and better understand how decisions in the SBI pipeline impact the resulting approximate posterior, we will first apply the SBI pipeline on a model where the ground truth posterior is known; namely an RC circuit model.

The equation for the RC circuit simulations is as follows:

$$C\frac{\mathrm{d}V}{\mathrm{dt}}(t) = \frac{E - V(t)}{R} + I_{\mathrm{e}}(t) \tag{5}$$

where $V(t)$ is the voltage response of an RC circuit to a current injection $I_{\mathrm{e}}(t)$. In our example we use a capacitance $C = 6$ F, resistance $R = 1\Omega$, and constant voltage source $E = 0$ V [1].

The example RC circuit simulations described here were parameterized in a way that will enable comparison to similar simulations in the biophysically detailed simulations described below for HNN (Fig 3C). More specifically, we drive the RC circuit with two square wave pulse injections, with positive and negative amplitudes lasting 20 ms each. Two parameters ($I_+$ and $I_-$) controlled the magnitude of positive and negative square pulse current injections, such that the sum $I_e = I_+ + I_-$ determined the final injected current for each time step. A third parameter, latency $\Delta t$, controlled the time delay between the two pulses. Specifically, $\Delta t$ shifts the negative current pulse in time, while the positive current pulse remains fixed. These inputs play a similar role as excitatory proximal and distal inputs in HNN (Fig 2). Table 1 details the prior distribution over the parameters used to generate training examples for SBI.

The relationship between current injection amplitude and the RC circuit voltage response strongly depends on if $\Delta t$ is zero or non-zero. When $\Delta t$ is zero, the total injected current sums to a single square pulse since the negative and positive pulses perfectly overlap. Fig 3A(top, blue) shows an example of this where a -0.2 mA square pulse current injection is delivered at 80 ms producing a voltage response with an exponential rise and decay with one peak. Since any combination of $I_+$ and $I_-$ which sum to -0.2 mA will produce an identical voltage, there will be an indeterminacy when attempting to infer these parameters from the voltage response. In contrast, Fig 3A(top, orange) shows an example with a non-zero latency $\Delta t \neq 0$. Specifically, the first current injection with $I_+ = 0.3$ mA is delivered at 80 ms, and the second current injection with $I_- = 0.5$ mA at 117.5 ms, therefore $\Delta t = 37.5$ ms. The voltage response exhibits two unique peaks due to the offset between the square pulses. Since there is only one combination of $I_+$ and $I_-$ that can produce this waveform, their values can be inferred exactly from the waveform. In other words, the amplitude parameters underlying the voltage response can be inferred exactly only when $\Delta t \neq 0$. Note that since we are approximating the posterior distribution, even values close to $\Delta t \approx 0$ will still produce an indeterminacy.

To visualize and interpret posterior distributions produced by SBI, we must first draw a sufficient number of random samples from the distribution. Since posterior distributions are often multidimensional, it is useful to plot the samples using a "pair-plot" (Fig 3B). The diagonal of the pair-plot is used to visualize the univariate (also known as marginal) distribution of each parameter, here using a kernel density estimate of $n = 1000$ posterior samples. The squares below the diagonal on the other hand visualize the bivariate distribution for pairs of

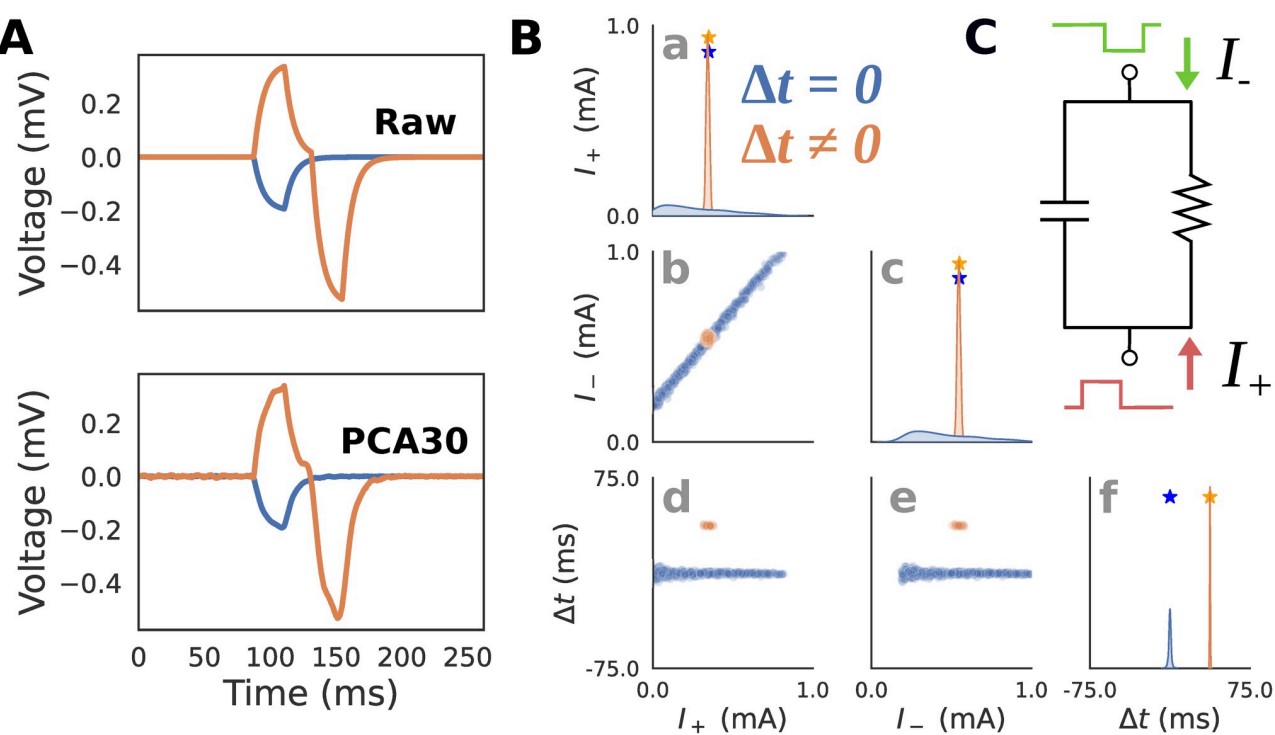

**Fig 3. RC circuit simulations. A**: Simulated voltage and current dipole waveforms are shown for two exemplar parameter configurations with latency between positive $I_+$ and negative $I_-$ square current injections, $\Delta t = 0$ (blue), and $\Delta t \neq 0$ (orange). An example of the original "Raw" waveform (top), as well as the PCA transformed waveform (bottom) with the first 30 components (PCA30) are shown to demonstrate that this summary statistic retains almost identical information. **B**: Posterior distributions showing the inferred values that can generate the blue and orange waveforms from panel A (PCA30 used to generate distributions) demonstrate that when the latency $\Delta t$ between the inputs is zero, their amplitudes are indeterminate as visible as a highly dispersed distribution on panels B(a-c, blue), and with a positive correlation between the parameters $I_+$ and $I_-$ on panel B(b, blue). In contrast, when $\Delta t \neq 0$ (orange), the distributions are tightly concentrated around the ground truth parameters (stars on panels B(a,c)) used to generate each simulation. The posterior distributions for the parameter $\Delta t$ are concentrated around the ground truth parameters for both conditions (panel B(f)). **C**: Schematic of how RC circuit is driven by positive $I_+$ and negative $I_-$ square current injections, where the amplitude and latency $\Delta t$ between pulses serve as parameters. This simulation parallels HNN simulations below in which a single excitatory proximal/distal input with variable synaptic conductances and latencies produce positive/negative deflections in the current dipole.

parameters by plotting the samples explicitly on a scatter plot. This example highlights that even with simple simulators, indeterminacies can easily arise necessitating the use of flexible posterior approximators (like masked autoregressive flows).

Note that PCA with 30 components (PCA30, variance explained = 0.883) was used as the summary statistic in this example (Fig 3A(bottom)) rather than the full time series to avoid the potential computational issues of conditioning posteriors on high dimensional data, while still retaining the majority of the waveform variance. Fig 3A (bottom) plots the inverse transformed PCA30 waveform to highlight that the summary statistic retains almost identical information. In all subsequent examples, PCA30 and PCA4 will refer to the $d = 4$ and $d = 30$ dimensional summary statistic vectors containing the loadings on the first 4 and 30 principle components. The PCA transformation was fit separately for each example using the dataset of 100,000 simulated time series waveforms (see Materials and methods for simulation details).

The results in Fig 3B show that the expected posterior distribution described above can be recovered with SBI. As shown in Fig 3B(a)–3B(e), when conditioned on the voltage response with $\Delta t = 0$, any distribution involving $I_+$ or $I_-$ (blue) will exhibit an indeterminacy (i.e., multiple recovered values along one dimension). The high correlation between $I_+$ and $I_-$ of 0.998 (p

**Table 1. Simulation parameters and SBI training.**

| RC Circuit | Range | Transform |
|---|---|---|
| Amplitude 1 (mA) | (0, 1) | linear |
| Amplitude 2 (mA) | (0, 1) | linear |
| Latency (ms) | (-75, 75) | linear |
| **HNN "RC"** | **Range** | **Transform** |
| Distal Exc (nS) | (1e-4, 1e-3) | exponential |
| Proximal Exc (nS) | (1e-4, 1e-3) | exponential |
| Latency (ms) | (-75, 75) | linear |
| **Beta Events** | **Range** | **Transform** |
| Distal Var (ms$^2$) | (0, 10) | linear |
| Proximal Var (ms$^2$) | (0, 40) | linear |
| **ERPs** | **Range** | **Transform** |
| Distal Exc (nS) | (1e-5, 1) | exponential |
| Proximal Exc (nS) | (1e-5, 1) | exponential |
| Distal Inh (nS) | (1e-5, 1) | exponential |
| Proximal Inh (nS) | (1e-5, 1) | exponential |

The prior support and transform function for the parameters of examples shown in the text.

$\approx 0$) demonstrates that the indeterminacy is characterized by a strong linear interaction between these parameters. Specifically, the line in Fig 3B(b, blue) corresponds to all values in which the amplitudes sum to a constant value of $V = -0.2$, and the resulting voltage waveform is identical. In contrast, the voltage response with $\Delta t \neq 0$ (orange) produces a posterior distribution concentrated on a single point around the ground truth parameters (Fig 3B(b, orange), correlation between $I_+$ and $I_-$: 0.375; $p < 1e-33$).

## Diagnostics enable comparison of posterior estimates using different summary statistics

In the previous example, we utilized PCA30 as a summary statistic to learn a low dimensional representation of the voltage time series. PCA is a common choice for dimensionality reduction, and has been used in historical MEG/EEG inference work with only the first 3-4 principal components [15, 42]. However, it is not guaranteed that PCA, which aims to only capture variance, will retain the features that best allow SBI to map waveforms to simulation parameters. An alternative approach is to leverage domain-expertise to construct hand-crafted summary statistics specific to the model and inference problem. Unfortunately, it cannot be known a priori which summary statistic will allow SBI to perform best, necessitating quantitative diagnostics that allow a systematic comparison. Here, we introduce two simple hand-crafted summary statistics, as well as the posterior diagnostics PRE and PPC, with the intention to build an intuitive understanding of how emphasizing different summary statistics can impact the final estimates produced by SBI.

The first hand-crafted summary statistic we defined is a four dimensional vector including the magnitude and timing of the maximum and minimum peaks (Peak) in the time domain of the simulated voltage response (Fig 4A(iii)). It can be readily seen that these features reflect the underlying simulation parameters. Upon visual inspection of the voltage response with $\Delta t \neq 0$ (orange), the height of the maximum and minimum peak directly correspond to the parameters $I_+$ and $I_-$, and the distance between these peaks correspond to the latency

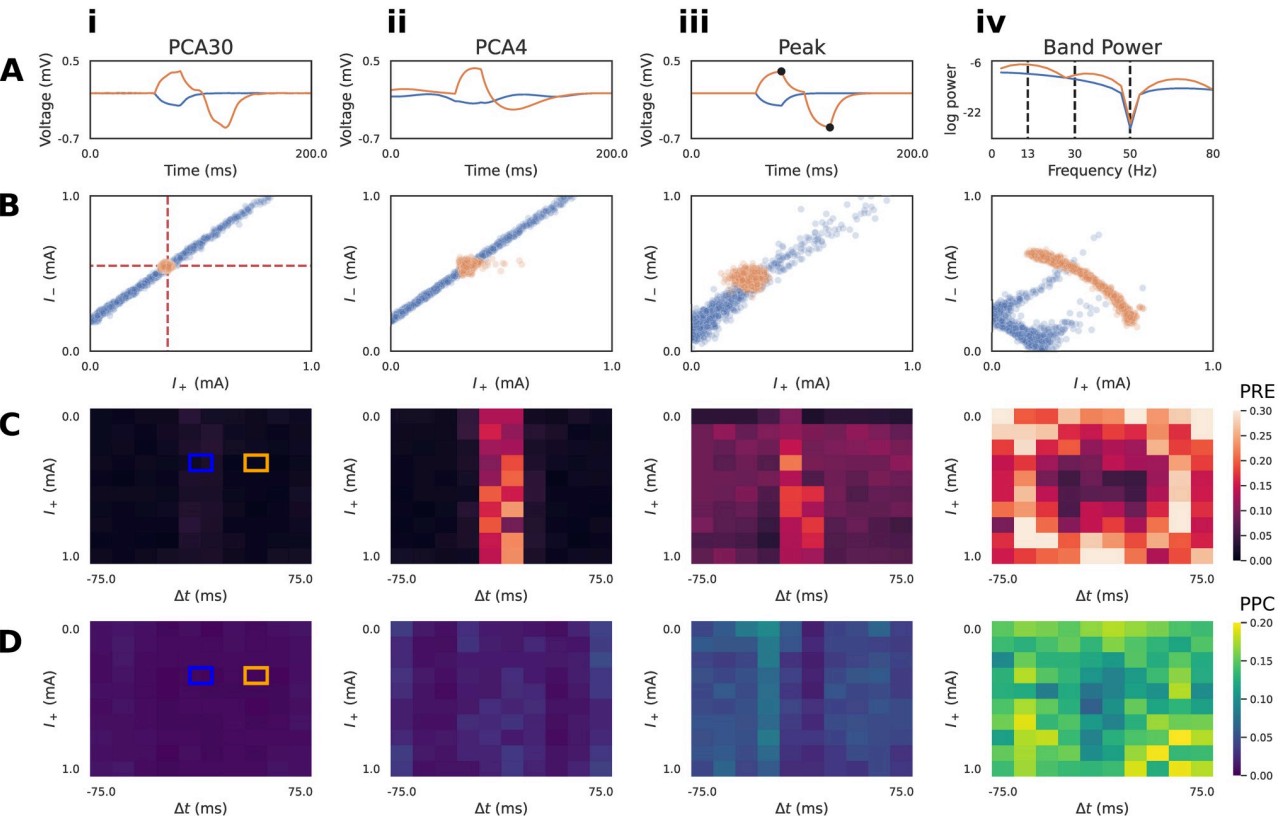

**Fig 4. Diagnostics to compare summary statistics on RC circuit model. A**: Summary statistics applied to the simulated time series included: `PCA30` (i), `PCA4` (ii), `Peak` (magnitude and timing of max/min, iii), and `BandPower` (four bands between dotted lines, iv). PCA plots (i-ii) show the associated inverse transformed signal. Two exemplar simulations with pulse latencies $\Delta t = 0$ (blue) and $\Delta t \neq 0$ (orange) are shown. **B**: Conditioning the approximate posterior distribution on the $\Delta t = 0$ time series produces indeterminacies for all summary statistics, such that the ground truth (red dotted lines) current injection amplitudes ($I_+$ and $I_-$) cannot be uniquely recovered. `PCA30`, `PCA4`, and `Peak` features exhibit a linear interaction between parameters for the $\Delta t = 0$ time series, whereas the ground truth is recovered for the $\Delta t \neq 0$ (orange) time series. `BandPower` produces non-linear interactions for both time series. **C**: Local parameter recovery error (PRE) heatmaps are shown. Brighter colors indicate higher dispersion of the posterior around the ground truth parameters defined by each square. Errors tend to be concentrated around $\Delta t = 0$ for `PCA30`, `PCA4`, and `Peak` features. **D**: Local posterior predictive check (PPC) heatmaps are shown. Brighter colors indicate regions where simulations generated from posterior samples are further from the conditioning time series. For both diagnostics, it is clear that `PCA30` produces the PRE and PPC across the parameter grid. The ground truths of the exemplar simulations of panels A/B are indicated by blue/orange squares.

parameter $\Delta t$. We'll show below that `Peak` features permits inference that is close to that achieved with `PCA30` Fig 4A(i) and also `PCA4` (variance explained = 0.610) Fig 4A(ii).

The second hand-crafted summary statistic we defined was a four dimensional vector including the band power (`BandPower`) of common frequency ranges used to study neural oscillations (Fig 4A(iv)). Specifically, we considered the beta (13-30 Hz), low-gamma (30-50 Hz), and high-gamma (50-80 Hz) ranges, as well as the aggregate band power of the alpha and lower frequency ranges (0-13 Hz). As we'll show below, this feature was intentionally selected as a cautionary example of a summary statistic that is ill-suited for the inference problem, but has a basis in previous neuroscience and Bayesian inference literature [47].

Next, we describe two diagnostics that allow comparison of desirable properties of the posterior distribution for different summary statistics, namely PRE and PPC (see Fig 1 Step 6 and Materials and methods). In the results below, we calculated PRE values over a grid, with 10 points for every parameter dimension, spanning the range of the prior distribution (Fig 4C).

Taking inspiration from [48], we visualize the PRE as heatmaps over the parameter grid (Fig 4C and 4D). One of the advantages of inspecting local posterior diagnostics ("local" as in specific to the pair $(x_0, \theta_0)$) is the ability to identify patterns. Fig 4C plots the PRE for the $I_+$ parameter, with respect to different ground truth values of $I_+$ itself, and the $\Delta t$ parameter. Blue and orange squares mark the ground truth values used to generate the waveforms in Fig 4A. We observe that the summary statistics PCA30, PCA4, and Peak all exhibit a pattern where $\Delta t$ values near zero produce a larger PRE compared to the rest of the parameter grid (Fig 4C (i)–4C(iii)). This is due to the indeterminacy in $I_+$ as seen in the blue posterior samples of Fig 4B(i)–4B(iii). It is apparent, however, that PCA30 values produce the lowest PRE values, even near a $\Delta t$ of zero. In contrast, the BandPower summary statistic produces a posterior distribution with complex indeterminacies for both observations. This results in high PRE values across the entire parameter grid (Fig 4C(iv)), indicating that this summary statistic is not effective at recovering the ground truth parameters.

The PPC is a method to describe how well samples from the posterior match the conditioning observation [34, 35]. Given a well-estimated posterior distribution $p(\theta \mid x)$ and a conditioning observation $x_0 \sim p(x \mid \theta_0)$, one would expect simulations $x_i \sim p(x_0 \mid \theta_i)$ to be close to the original conditioning observation. Unlike the PRE heatmaps, the PPC plots shown in Fig 4D does not exhibit obvious patterns with respect to the underlying parameter grid.

These diagnostics demonstrate PCA30 performs the best for this inference problem. Additionally, the local PRE analysis revealed differences between summary statistics that were not apparent with the global diagnostics. It is important to note that neither of these diagnostics quantify the closeness of the approximation $\Phi$ to the true posterior. For instance, if there is an interaction between parameters of the model causing a parameter indeterminacy, then the PRE will always be non-zero, since the posterior distribution will be spread in the parameter space. This is the case for the posterior of the RC circuit with $\Delta t = 0$ in Fig 3B. Similarly, if model simulations are stochastic, then a given set of parameters may map to multiple equally valid outputs, producing a non-zero PPC. Nevertheless, both diagnostics provide useful information to compare desirable properties of the posterior approximation.

## Subthreshold HNN simulations mimicking the RC circuit shows that inference with summary statistics that account for the full time series waveform perform best

Building from the RC circuit, we now describe a nearly identical inference problem in our large-scale biophysically detailed model constructed to study the neural mechanisms of human MEG/EEG, HNN (Fig 5). As described further in Materials and methods, HNN is a neocortical column model with multiple layers and biophysically detailed local cell types. The local network receives exogenous excitatory synaptic input through layer specific pathways that effectively synapse on the proximal and distal dendrites of the pyramidal neurons, representing "feedforward" and "feedback" input from thalamus and higher order cortical areas. These inputs are simulated with external "spikes" that activate layer specific excitatory synapses (see Fig 2B and 2C, and reduced schematic in Fig 5C). This synaptic activity induces current flow within the pyramidal neuron dendrites, which is summed across the population to simulate a net current dipole that is directly comparable to that measured with MEG/EEG. Several previous studies have shown that patterns of activation of the local network through these pathways can generate commonly measured MEG/EEG current dipole signals such as event related potentials and low frequency brain rhythms, e.g., see [5].

To set up an inference problem that is comparable to our RC circuit example, we begin by considering patterns of drive to the network that create subthreshold current flow in the

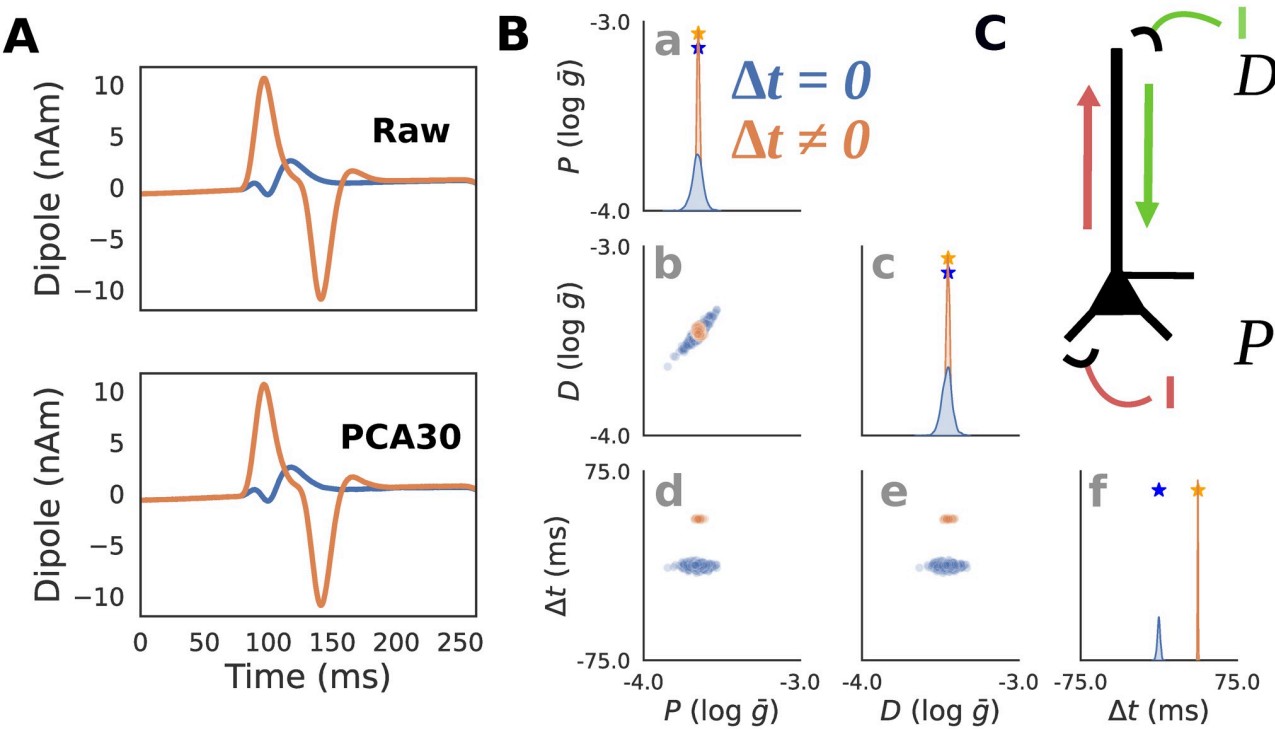

**Fig 5. HNN simulations that mimic RC circuit.** HNN simulations that reflect the nearly identical parameter configuration as the RC circuit in Fig 3.
**A**: Simulated current dipole waveforms are shown for two exemplar parameter configurations with $\Delta t = 0$ (blue) and $\Delta t \neq 0$ (orange). The original "Raw" simulated waveform (top) is plotted in comparison with the PCA inverse transformed waveform with 30 components (PCA30, bottom). **B**: Posterior distributions showing the inferred values that can generate the waveforms from panel A demonstrate that when the latency between the inputs is zero (blue), their amplitudes are indeterminate as visible as a dispersed distribution on panels B(a-c, blue), with a positive correlation between the parameters $P$ and $D$ on panel B(b, blue). Unlike the previous example (Fig 3), the indeterminacy is notably smaller for $\Delta t = 0$, with the posterior distributions primarily being concentrated around the ground truth parameters for $P$ and $D$ (stars on panels B(a,c)). **C**: Schematic of HNN simulations in which a single excitatory proximal/distal input with variable synaptic conductances and latencies produce positive (red)/negative (green) deflections in the current dipole.

pyramidal neurons, effectively "disconnecting" the network, because local synaptic interactions depend on local cell firing. Simulations with spiking dynamics will be demonstrated in the following section. For simplicity, we first describe the subthreshold current flow in the L5 pyramidal cells only (Fig 5C). Specifically, we ran HNN simulations with single exogenous spikes that activate excitatory synapses on the proximal and distal dendrites of L5 pyramidal cells. Synaptic excitation of distal synapses generates current flow down the dendrites (e.g. see green arrow Fig 5C), and excitation of proximal dendrites generates current flow up the dendrites (e.g. see red arrow Fig 5C). A delay between these the time of the two driving spikes can create a net current dipole signal that is analogous to that observed in the RC circuit for a non-zero time delay between the applied currents (see Fig 5A, orange). Further, when the delay between the spikes is zero (see Fig 5A, blue) an indeterminacy in parameter estimation can occur, as described below.

With this set up, we applied SBI to infer parameters that mimic those of the RC circuit example, using PCA30 (variance explained = 0.976) as the chosen summary statistic to constrain the inference problem. Namely, the strength of proximal and distal excitatory inputs, referred to as $P$ and $D$, parameterized as the maximal conductance at their respective synapses and the latency $\Delta t$ between the two inputs. We kept the proximal input time fixed, and let the

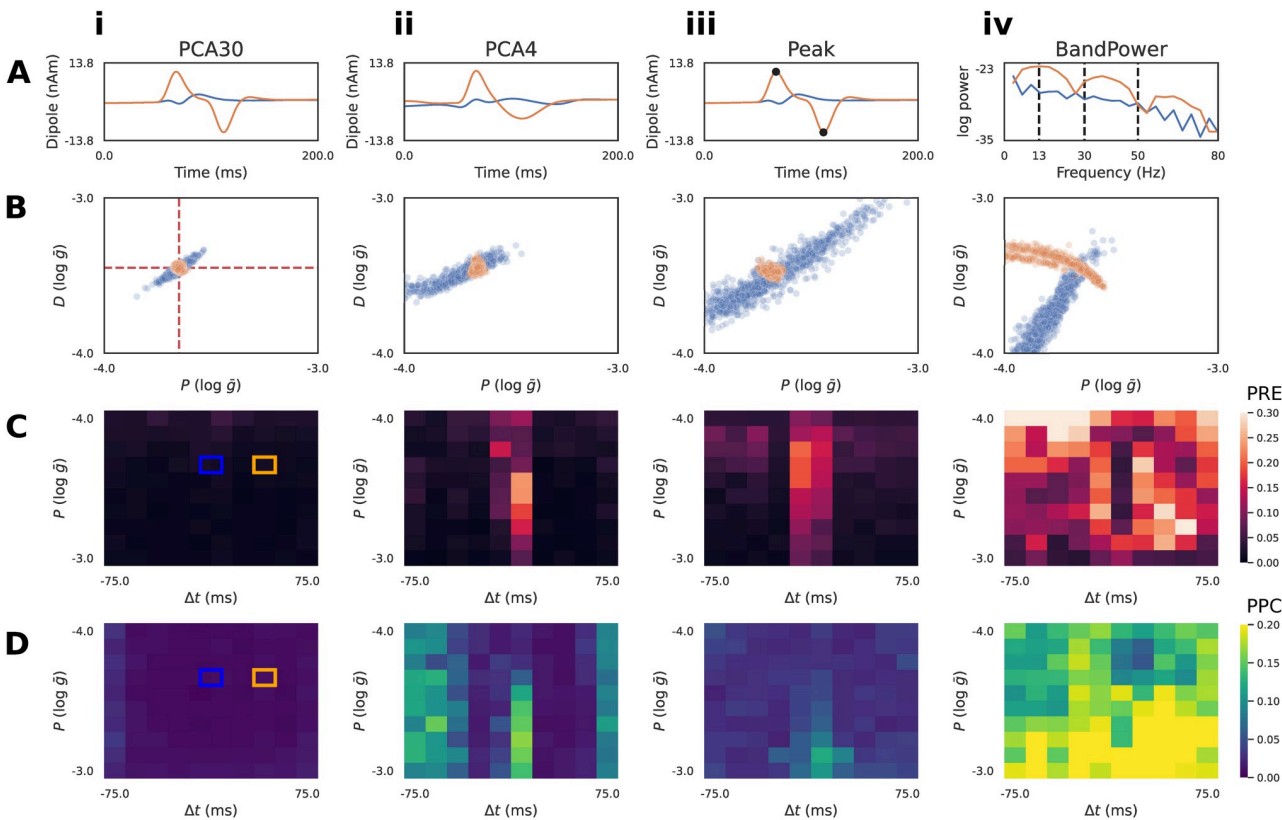

**Fig 6. SBI diagnostics of summary statistics in HNN.** The analysis shown in Fig 4 is repeated on a simplified HNN simulations for comparison. **A**: Summary statistics included PCA30 (i), PCA4 (ii), Peak (iii), and BandPower (iv). Two exemplar simulations with input latencies $\Delta t = 0$ (blue) and $\Delta t \neq 0$ (orange) are shown. **B**: We show the approximate posterior when conditioned on both the exemplar waveforms. The $\Delta t = 0$ time series produces a positive correlation between $P$ and $D$ for all summary statistics. **C**: Local parameter recovery error (PRE) is shown. Unlike the RC circuit, PCA30 and PCA4 permit better ground truth recovery even when $\Delta t$ is near zero. In contrast, Peak features have poor parameter recovery similar to the RC example **D**: Local posterior predictive checks (PPC) are shown. PCA30 and PCA4 produce the values across the parameter grid.

distal input time vary with $\Delta t$. The prior distribution over $P$ and $D$ was set to ensure that all simulations remained subthreshold (see Table 1 for prior distribution ranges).

Similar to the RC circuit example, simulations with $\Delta t = 0$ produce a current dipole with a reduced magnitude (Fig 5A(blue)), whereas $\Delta t \neq 0$ produces a clear positive and negative peak (orange). As shown in Fig 5B(a-e, blue), when conditioned with $\Delta t = 0$, any posterior distribution involving $P$ and $D$ will exhibit an indeterminacy. Intuitively, this indicates that the proximal and distal inputs can compensate within a small range to produce similar current dipole waveforms. Unlike the RC circuit, this interaction does not span the full range of input strengths, and instead is more tightly concentrated around the ground truth (Fig 5B(a), 5B(c) and 5B(f) stars).

Once again, we show that the choice of summary statistics impact the learned posterior distribution approximation, and that diagnostics can be used to evaluate the quality of the parameter estimation (Fig 6A and 6B). When $\Delta t \neq 0$, both PCA30 and PCA4 (variance explained = 0.665) produced a posterior that is localized around the ground truth (Fig 6B(i) and 6B(ii), orange). When $\Delta t = 0$, the posterior was still concentrated but with a slight indeterminacy (Fig 6B(i) and 6B(ii), blue) that was less prominent than the analogous simulation in the RC circuit (Fig 4B(i) and 4B(ii), blue). The Peak summary statistic produced a posterior

that is well clustered around the ground truth for $\Delta t \neq 0$ (Fig 6B(iii), orange), but for $\Delta t = 0$ exhibited a much more striking indeterminacy (Fig 6B(iii), blue). In contrast, BandPower was insufficient for ground truth recovery for both the $\Delta t \neq 0$ and $\Delta t = 0$ as was the case for the RC circuit simulations (Fig 6B(iv)).

We quantified desirable properties of the posterior estimates, with the local PRE and PPC analysis described above. At $\Delta t \approx 0$, PRE values were large when using BandPower features (Fig 6C(iv)), relatively smaller for PCA4 and Peak (Fig 6C(ii) and 6C(iii)), and almost completely disappears for PCA30 (Fig 6C(i)). The PPC values for the BandPower show that inference using this summary statistic produced results that were highly dissimilar to the conditioning observations (Fig 6D(iv)), while PCA30, PCA4, and Peak exhibit a much lower PPC values (Fig 6D(i)–6D(iii)). Local PRE and PPC analysis largely agrees with the summary statistic performance captured by the RC circuit PPC heatmaps (Fig 4C and 4D).

In summary, intelligently chosen summary statistics like Peak features can perform well, but leveraging information from the entire time series using PCA30 produced consistently lower PRE and PPC values. The highly effective parameter recovery across the entire parameter grid for PCA30 suggests that SBI permits a near unique mapping from dipole waveform to parameters when the summary statistic accounts for the full time series waveform and the parameters are kept in a subthreshold regime. In the next example, we show that this is not true in general. Even when using information from the full dipole waveform with PCA, inference in HNN simulations that include stochasticity can produce substantial parameter indeterminacies.

## SBI expands previously proposed mechanisms of subthreshold Beta Event simulations in HNN and shows stochastic simulation can lead to indeterminancies

Previous studies have applied the HNN modeling framework to propose novel mechanisms for the cellular and circuit level generation of transient low frequency oscillations in the 15-29 Hz beta-frequency band, referred to as Beta Events or beta bursts [24, 28, 49]. Many studies have shown that Beta Events occur throughout the brain (e.g. see [50]) and their expression correlates with healthy and pathological sensory and motor processing [38, 49, 51]. Further, they often have a stereotypical waveform shape that resembles an inverted Ricker wavelet lasting ~150 ms [24, 49, 51–53] (see Fig 7). HNN provides potential mechanistic explanations for how the Beta Event waveform is generated and how changes in waveform shape may emerge. The HNN derived novel Beta Event mechanism put forth by Sherman et al. [24] showed that Beta Events can arise from the dendritic integration of coincident bursts of subthreshold proximal and distal dendritic excitatory synaptic inputs to cortical pyramidal neurons [24, 28], such that the distal drive is effectively stronger and lasts one beta period (~50 ms); a prediction that was supported by invasive laminar recordings in mice and monkeys [24] and high-resolution MEG in humans [49]. More specifically, HNN reproduced Beta Events with the observed stereotypical shape when stochastic trains of action potentials were simulated to drive the proximal and distal dendrites of the pyramidal neurons, nearly simultaneously. Bursts of input whose mean timing and standard deviation where chosen from Gaussian distributions activated excitatory synapses in a proximal and distal connection pattern, as shown in Fig 7A. The inverted Ricker waveform shape depended on the standard deviation of the proximal burst being broader than the distal burst, with the proximal burst occurring over ~150 ms and the distal burst occurring over ~50 ms, and the mean time of each burst being the same, i.e. reminiscent of $\Delta t = 0$ above. The proximal drive pushes current flow up the pyramidal neuron dendrites, while the distal drive pushes it back down (see Fig 7A).

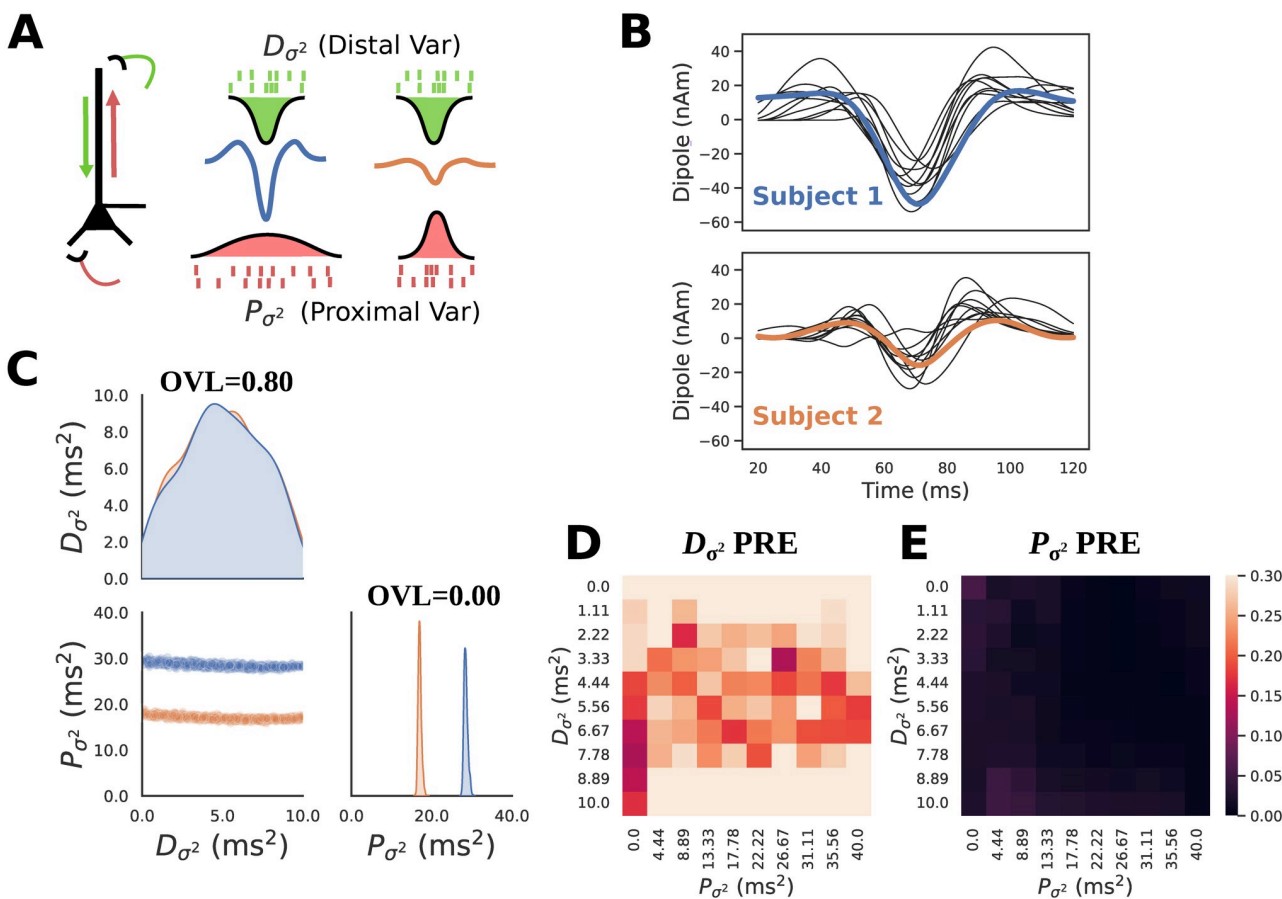

**Fig 7. HNN-SBI recovers circuit mechanism of Beta Event magnitude described in previous studies. A**: Schematic of Beta Event simulations in HNN. Beta Events are generated by a simultaneous burst of subthreshold proximal (red) and distal (green) excitatory inputs to L5 pyramidal neurons. **B**: Average source localized MEG Beta Event waveforms recorded from two subjects. Subject 1 (top, blue) exhibits a larger magnitude trough compared to subject 2 (bottom, orange). Simulations corresponding to a posterior predictive check (PPC) are shown in black, such that the parameters were sampled from the posterior (panel C) of each respective waveform. **C**: Posterior distributions conditioned on large (blue) and small (orange) magnitude Beta Events demonstrate that larger proximal variance produces a larger magnitude trough. Overlap coefficients (OVL) quantifying the separability of the marginal posterior distributions conditioned on each waveform are shown on the diagonal for the corresponding parameters. The marginal distributions for distal variance are highly overlapping, and non-overlapping for the proximal variance **D**: PRE heatmap of $D_{\sigma^2}$ shows accurate parameter recovery (dark colors) when the ground truth parameters of $D_{\sigma^2}$ are around 5 ms², and quickly worsen (light colors) as $D_{\sigma^2}$ increases or decreases. **E**: PRE heatmap of $P_{\sigma^2}$ shows accurate parameter recovery across the entire range of the prior distribution for both $P_{\sigma^2}$ and $D_{\sigma^2}$.

Previous work also showed that, when the parameters of the proximal drive were held constant, the magnitude of the prominent middle trough changed with the variance of the distal drive, such that a parametric lowering of the variance pushed more current flow down the dendrites generating an increased magnitude and sharper trough [24]. This previous study led to the hypothesis that variance of the inputs impacted the waveform shape but was limited in that it did not perform an automated parameter inference on empirical data, nor did it investigate the additional role of proximal drive variance. Extending these prior results, here we hypothesized that both proximal and distal drive variances can impact the waveform shape and thus apply the SBI methods to HNN to infer distributions of proximal and distal drive variance in empirical MEG data from two example subject's whose average Beta Event trough magnitudes are distinct, collected in the prior study (see Methods). In this example, Subject 1 (Fig 7B

(blue)) had larger magnitude peaks and troughs compared to subject 2 (Fig 7B(orange)). The same diagnostics used above are applied to assess the quality of the estimates.

We used the same parameters for Beta Event generation as in [24] and defined a prior distribution over proximal and distal input variance ($P_{\sigma^2}$, $D_{\sigma^2}$), see Table 1. All other parameters, including the number of spikes and mean input time, were held fixed as in [24]. We used PCA30 (variance explained = 0.998) as the summary statistic motivated from the results above. We ran the HNN-SBI workflow to obtain a posterior distribution approximation that allows us to infer $P_{\sigma^2}$ and $D_{\sigma^2}$ for a given waveform (Fig 7B and 7C).

For both small and large magnitude Beta Events, there is a clear indeterminacy in $D_{\sigma^2}$. such that the distribution of $D_{\sigma^2}$ is widely spread over the range of the prior distribution, indicating high uncertainty in the parameter estimates (Fig 7C). In contrast $P_{\sigma^2}$ exhibits highly concentrated distributions, with the larger magnitude (blue) Beta Event corresponding to larger values ($P_{\sigma^2} \approx 30$ ms$^2$), and the smaller magnitude (orange) Beta Event corresponding to smaller values ($P_{\sigma^2} \approx 18$ ms$^2$). With a large proximal variance and smaller distal variance, the distal drive is effectively stronger and able to counteract the upward current flow to create a large downward deflection. With a small proximal variance, the stronger upward current flow is more equally matched with the downward current flow to produce an overall smaller and noisier downward deflection. This proof-of-concept example based on two subjects is not meant to make strong scientific predictions on parameters controlling Beta Event shapes, but exemplifies how the HNN-SBI framework can be applied to real data.

To quantify the separability of these distributions, we can employ the distribution overlap coefficient (OVL) which varies on a scale of [0, 1] such that 1 indicates complete overlap, and 0 indicates no overlap. Unsurprisingly, the $D_{\sigma^2}$ distributions for the large and small Beta Events produce an OVL of 0.80 due to the clearly visible high degree of overlap, whereas when comparing the $D_{\sigma^2}$ distributions they exhibit almost no overlap with an OVL of approximately 0.00 (Fig 7C).

We note that unlike the simulations from the previous sections, where all parameters were deterministic, the exogenous drive times considered here are stochastic. As a result, there is not a 1:1 mapping from parameters to simulation output. To highlight this, Fig 7B shows simulations with parameters drawn from their respective posterior distributions (black traces). This visually represents the PPC diagnostic, where simulations from each posterior are close to the conditioning observation, but not a perfect match. In this setting, the stochasticity in the simulations make it so that measures such as PPC's are not guaranteed to be zero even with a perfect approximation of the ground truth posterior distribution. We additionally observe a pattern in the local PRE heatmap of $D_{\sigma^2}$, indicating that this parameter is accurately recovered from simulations generated with a $D_{\sigma^2}$ of approximately 5 ms$^2$ (Fig 7D) and worsens for higher and lower values of $D_{\sigma^2}$. The PRE heatmap of $P_{\sigma^2}$ shows all values close to zero demonstrating that the parameter is accurately recovered across the entire range of the prior distribution (Fig 7E).

In summary, the Beta Event example demonstrates how SBI can be used to estimate parameter distributions for given time series waveforms, and to compare potential mechanisms underlying different waveforms. For the example comparison shown, the variance of the proximal drive could be uniquely inferred while the variance of distal drive could not.

## SBI reveals parameter interdependencies for suprathreshold event related potential simulations in HNN

In the Beta Event example above, the effective strength of the proximal and distal input were maintained in a range where the activity of the cells remained subthreshold, which naturally

limits the dynamic range of the simulation. Next, we consider a more complex example, in which the cells are driven to a suprathreshold spiking regime and show that this additional complexity can lead to parameter estimation indeterminacy that indicates a compensatory interaction between parameters.

The example considered describes simulations of a sensory evoked response or event related potential (ERP). HNN has been applied to study source localized ERPs in primary somatosensory cortex from tactile stimuli (e.g. [25, 26]) and in primary auditory cortex from auditory stimuli (e.g. [27]). In both cases, ERPs were simulated with a sequence of exogenous input that represented an evoked volley of drive to the local circuit that occurs after a sensory stimulus. The drive sequence consisted of a proximal drive representing the initial feedforward input from the thalamus, followed by a distal drive representing feedback input from higher order cortex, followed by a second proximal drive representing a loop of re-emergent thalamo-cortical drive (see schematic red and green arrows in Fig 8A. These drives were strong enough to generate spiking interactions in the local network and induced current flow up and down the pyramidal neuron dendrites to generate a current dipole ERP waveform analogous to those experimentally recorded (Fig 8A). Note, here we are not examining recorded data, but only example simulations. The specific timing of this exogenous drive sequence for example simulations is shown with arrows in Fig 8B. The parameters regulating the timing and the strength of these drives were fixed to the same values for the different conditions considered.

HNN has also been applied to infer neural mechanisms underlying differences in ERP waveform shapes recorded across different experimental conditions (e.g, for tactile evoked responses in [25]). Here, we are not trying to reproduce any empirical findings but rather apply SBI to an ERP simulation as in [25] to examine the influence of changes in local network connectivity on the ERP waveform as a proof of concept example that examines a small subset of parameters distinct from our prior investigation.

We start by simulating example ERPs with different peak magnitudes as shown in Fig 8B as follows. ERPs were generated using a sequence of exogenous proximal-distal-proximal inputs (Fig 8A and as described above). The parameters representing the timing and strength of the sequence of exogenous proximal and distal input to the local circuit were chosen to be those distributed with the HNN software representing an example tactile evoked response simulation and fixed to those values (see Table 1). We then defined a prior distribution over parameters that define a small subset of the local network excitatory and inhibitory connectivity Fig 2A. These parameters included the maximum conductance ($\bar{g}$) of layers 2 and 5 excitatory AMPA ($E_{L2}$ / $E_{L5}$) and inhibitory GABA$_A$ ($I_{L2}$ / $I_{L5}$) connections to the L5 pyramidal cell. Specifically, $E_{L2}$ and $I_{L2}$ pertains to synapses on the distal dendrites, $E_{L5}$ on proximal dendrites, and $I_{L5}$ on the soma. Note that there exist more local network connections than those varied here as shown in Fig 2A and this chosen prior distribution was not based on any hypothesis or prior knowledge of the impact of local parameters on the ERP, but rather as a tractable example.

Two example ERPs produced by networks with different local connectivity are shown in Fig 8B. The ground truth parameters that created these waveforms are shown with stars in Fig 8D. Despite being activated by an identical exogenous input sequence, it is apparent that the local network connectivity differences lead to dramatically distinct current dipole ERP waveforms and corresponding spiking activities. In Fig 8C the spiking activity associated with each waveform is largely distinguished by the activity of L5 pyramidal neurons (red dots), with more firing in Condition 2 (orange waveform). For Condition 1 (blue), the first proximal input leads to the beginning of a sustained negative deflection in the current dipole, which persists during the subsequent distal input due to prolonged activation of the L5 basket cells which inhibit the L5 pyramidal neuron soma to pull current flow down the dendrites. Once

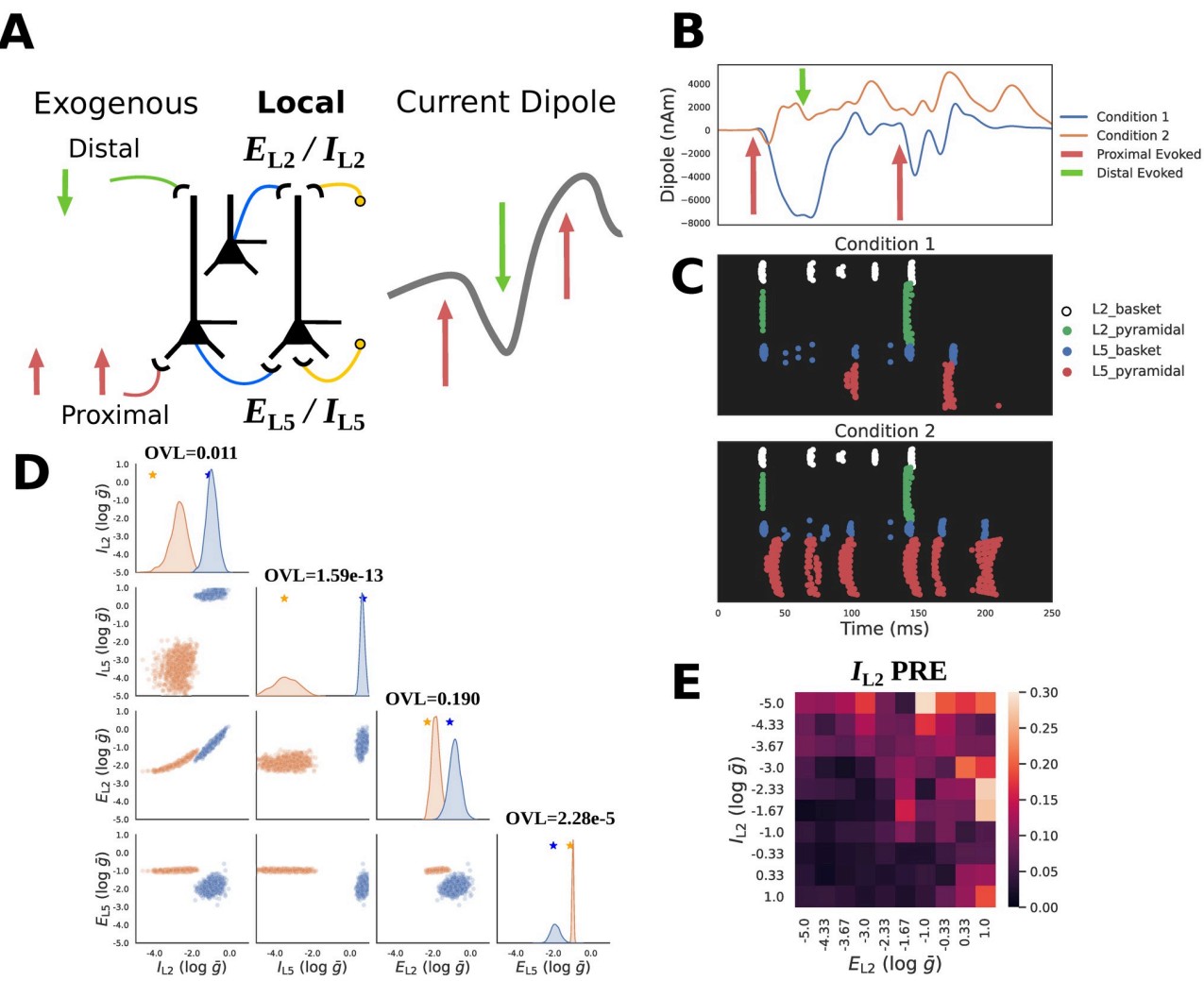

**Fig 8. Inferring local connectivity parameters from ERP waveforms. A**: Schematic of ERP simulations in HNN. Evoked activity is driven by a fixed sequence of proximal-distal-proximal exogenous inputs. SBI is used to infer the maximal conductance strength ($\bar{g}$) of local excitatory/inhibitory connections to the proximal/distal dendrites of L5 pyramidal neurons for example waveforms. **B**: Exemplar simulated ERPs (blue and orange solid lines) with differing local connectivity strengths chosen from a defined prior distribution (described in the text) are shown, along with the fixed timing of the sequence of exogenous inputs for each simulation (red and green arrows). **C**: Spike raster plots of cell specific firing for the two ERP simulation conditions from panel B. **D**: Posterior distributions over local connectivity parameters alongside ground truth parameters (stars on diagonal) for conditioning observations. A strong interaction between excitatory/inhibitory distal inputs ($E_{L2}$ and $I_{L2}$) is visible in the lower square. Overlap coefficients (OVL) quantifying the separability of the marginal posterior distributions conditioned on each waveform are shown on the diagonal for the corresponding parameters. $E_{L2}$ and $I_{L2}$ exhibit a small amount of overlap with OVL values of 0.011 and 0.190 respectively. In contrast $E_{L5}$ and $I_{L5}$ were much more distinguishable, exhibiting OVL values of 2.28e-5 and 1.59e-13 respectively. **E**: Local parameter recovery error (PRE) for distal inhibition $I_{L2}$ indicates errors are higher for observations generated with strong excitatory $E_{L2}$ and weak inhibitory $I_{L2}$ distal connections.

this inhibition ends, the L5 pyramidal neuron is able to spike, pushing current flow back up the dendrites and the subsequent volley of proximal drive continues to push current flow up and down the pyramidal neuron dendrites due to a similar spiking dynamic. This is in contrast to Condition 2 (orange) in which L5 pyramidal neuron spiking starts almost immediately after the first proximal input and persists, pushing current flow up the dendrites to create a sustained positive deflection in the current dipole that persists through the entire simulation.

Fig 8D shows the results of applying the HNN-SBI framework with the PCA30 (variance explained = 0.999) summary statistic to estimate the ground truth parameters that generated

the ERP waveforms described above. It is apparent that the posterior distributions conditioned on each waveform place high probability mass around the corresponding ground truth parameters (Fig 8D, stars on diagonal), but also exhibit strong indeterminacies. For example, for each condition, there is a clear interaction between $E_{L2}$ and $I_{L2}$, such that as one parameter increases the other also increases, suggesting that these two parameters can compensate one another in a limited range to maintain a constant waveform. We can also observe that between the two conditions there are apparent differences in L5 pyramidal neuron spiking, as well as the sustained negativity (blue) and positivity (orange) observed in the current dipole due to the complex dynamics that each parameter configuration creates.

To quantify the separability of these distributions, we calculated the OVL coefficient for the marginal distributions of all parameters (Fig 8D diagonal). Estimated parameter distributions of the synapses on the layer 5 distal dendrites, $E_{L2}$ and $I_{L2}$, exhibit a small amount of overlap across conditions with OVL values of 0.190 and 0.011 respectively. The parameter distribution of the synapses on layer 5 somas however were much more distinguishable for the two conditions, exhibiting OVL values of 2.28e-5 for $E_{L5}$, and 1.59e-13 for $I_{L5}$. We additionally performed diagnostic analysis of the local PRE to see how recovery of $I_{L2}$ changes as a function of $I_{L2}$ itself, and $E_{L2}$. Fig 8E exhibits a clear pattern indicating that the recovery of $I_{L2}$ is worse when the network exhibits low $I_{L2}$ and large $E_{L2}$.

In summary, the ERP example provides another demonstration of how SBI can be used to estimate parameter distributions for given time series waveforms and to compare potential mechanisms underlying different waveforms. For the hypothetical example comparison shown, $E_{L5}$ and $I_{L5}$ were uniquely inferred with distributions for the waveforms compared exhibiting very little overlap. Distributions for $E_{L2}$ and $I_{L2}$ were also separable, albeit with slightly more overlap, and a marked interaction between these two parameters.

## Discussion

Inference in detailed biophysical models like HNN is far from a solved problem. Nevertheless, the recent developments in likelihood-free inference techniques have enabled predictions of parameter distributions with a level of detail and complexity that was simply not feasible until now. In this study, we detailed step-by-step methods to employ SBI in detailed models of neural time series data and to assess the quality and uniqueness of the estimated parameter distributions. We began with a simplified RC-circuit example which exemplified the possibility of parameter interactions and highlighted limitations with chosen summary statistics that do not consider the full time series waveform. We then demonstrated how distributions of biophysical parameters that can account for a given time series waveform can be inferred using an integrated HNN-SBI workflow applied to two common MEG/EEG signal motifs (Beta Events and ERPs) and how to assess overlap of the distributions from two different waveforms. This work does not aim to be an exhaustive guide to inference in HNN, nor to focus on specific neuroscientific questions, but instead to provide useful examples and methods to highlight critical decisions in the inference pipeline. There are several major takeaways from our study. First, highly non-linear biophysically detailed neural models like HNN are not suitable for Bayesian estimation methods that require access to a likelihood function (e.g. variational inference) or that approximate posterior distributions with independent Gaussians (a.k.a Laplace approximation). Rather they necessitate a method that can estimate complex posterior distributions and parameter interactions from a simulated dataset (e.g. SBI with masked autoregressive flows). Second, an important initial step in the SBI process is to identify a limited set of parameters and a range of values for those parameters that are assumed to be able to generate the waveform of interest and variation around it (i.e. the prior distribution). Due to the large-scale

nature of biophysically detailed network models, it is not possible to perform SBI on all parameters at once. The choice of the prior distribution represents a hypothesis about parameters that are assumed to contribute to variation in the waveform. This hypothesis can be informed by domain knowledge of the question of interest. In the HNN examples shown, the hypothesized parameters of interest for estimation were the strength of the proximal and distal excitatory synaptic drive for the Beta Event simulation, and local excitatory and inhibitory connectivity for the ERP simulation; these parameters were chosen only for didactic purposes. All other parameters were fixed based on previous studies. Third, posterior diagnostics like PRE and PPC, are valuable tools to guide decisions in the inference pipeline, e.g. optimal summary statistic selection, and OVL can be used to assess the uniqueness of estimated distribution for two different waveforms. Fourth, when estimating parameters that account for time series waveforms, summary statistics informed by the full time series such as PCA are the most effective at retaining essential information for mapping recorded signals to underlying parameters. While hand-crafted summary statistics, such as peak latency or magnitude, can permit an accurate mapping for certain waveform features, their selection may be insufficient to identify unique parameters distributions.

## Comparison with inference in MEG/EEG neural modeling frameworks that rely on dynamic causal modeling

To our knowledge, while there are several modeling frameworks for simulating MEG/EEG signals, the other frameworks that use likelihood-based Bayesian inference to estimate parameters fall in the category of Dynamic Causal Modeling (DCM) [15, 54]. It is important to emphasize that while the HNN-SBI framework conceptually overlaps with DCM, they are two fundamentally distinct techniques which address different questions. At its base, DCM combines variational inference, a computationally efficient Bayesian inference algorithm, with neural mass models, as well as an observation model which translates simulated activity to experimental measures (e.g., MEG/EEG). Neural mass models refer to a specific class of neural models where a single variable represents the aggregate activity of large neural populations (e.g. population spike rates). The inferred parameters in the DCM framework most often represent the coupling strength between distinct neural masses (e.g. population nodes). By making simplifying anatomical and physiologic assumptions, neural mass models in the DCM framework can be employed in a large variety of inference problems due to their computational efficiency. However, their ability to make precise biophysical predictions on cellular and local circuit level processes is limited as the parameters are an abstraction representing population level activity [54], preventing a one-to-one comparison between model predictions and experimental measurements. Further, physiologically important effects like dendritic backpropagation are not represented in neural mass models unlike HNN. There are, however, advantages of DCM over the HNN-SBI framework that access to a known likelihood function and other simplifying assumptions allow. For example, one critical question that the HNN-SBI framework is currently not suited to address is inference with multiple spatially separated cortical sources. While theoretically possible, the high computational demands of HNN-SBI severely limit the ability to explore multi-area interactions, and highlight the importance of using neural mass models and DCM in the analysis of whole-brain neuroimaging data. Recent work has shown that neural mass models can also be integrated with the SBI-framework for whole-brain studies [23], highlighting the adaptability of the SBI methodology to a wide variety of neural models. Another major limitation of the current framework is that a small number of parameters must be chosen to create a sufficiently large dataset for amortized inference. Recent methodological developments in Bayesian model reduction with DCM have shown that large parameter

spaces can be searched over in a computationally efficient manner [55]. Alternatively, previous works have attempted to merge the DCM and HNN modeling frameworks [56, 57]. This approach potentially enables one to benefit from the computationally efficient inference procedures offered by DCM, as well as the biophysical interpretability offered by HNN. Importantly, this approach necessitated 1) grouping of biophysical parameters in HNN to enable a one-to-one mapping to DCM parameters, and 2) an equivalent number of excitatory and inhibitory neurons in each cortical layer to permit a mean-field approximation and was applied to infer parameters in non-spiking HNN simulations of brain rhythms. For hypotheses where these assumptions apply and precise biophysical detail and network heterogeneity is not a focus the HNN-DCM approach offers a compelling alternative to HNN-SBI inference framework.

## Comparison to other biophysically detailed neural modeling studies and estimation techniques

The simulation process outlined here extends prior work using SBI to estimate parameters in detailed neural models. Prior work applying SBI to neural models has included a single compartment Hodgkin-Huxley model, and the stomatogastric ganglion model [18], both of which include an extensive parameter set, but contain significantly less detail and are smaller scale models than HNN. Additionally, non-amortized inference was performed in these models using sequential neural posterior estimation, allowing a much larger parameter set to be inferred, but only for a specific single observation. In contrast, we omit the use of sequential methods to perform inference on multiple observations using the same trained neural density estimator, but at the expense of the number of parameters that can be inferred simultaneously. Nevertheless, when only a few parameters are necessary for estimation, SBI offers a powerful tool for amortized inference in high-dimensional models, with examples ranging from whole-brain models in systems neuroscience [23, 58], to particle physics models investigating the formation of the Higgs boson [59]. The SBI workflow applied here used the neural density estimation technique known as masked autoregressive flows. There are currently a large number of neural density estimation techniques beyond this choice, each offering distinct advantages such as sample efficiency, expressivity, and likelihood evaluation [16]. While this field is rapidly evolving, recent concerns have been raised about the limits of such tools in the domain of Bayesian inference for scientific discovery [44, 60]. Unfortunately, there currently exist very few techniques for the validation of posterior distributions learned through neural density estimation beyond PPC and PRE diagnostics shown here. One promising work is simulation-based calibration [61], which plays a similar role as PPC's by measuring properties the posterior approximation should satisfy if it is close to the ground truth. It is important to note, however, that this technique assesses the quality of the posterior approximation for the marginals of each parameter separately. More research in the domain of multidimensional calibration in the context of likelihood-free inference will be crucial to better represent complex parameter interactions like local E/I balance, as shown in Fig 8. Nearly all parameter estimation techniques in high-dimensional biophysically detailed neural modeling will be computationally expensive. Indeed, SBI with HNN has a high computational load (for each example shown here, 100,000 simulations were run on a computing cluster in parallel over 512 CPU cores). While the upfront computational costs are high, there are advantages to SBI over other estimation techniques, for example COBYLA estimation, which has also been applied in HNN [5]. The main distinguishing factor is that SBI makes use of every simulation to build an accurate approximation of the posterior distribution for many waveforms. In contrast, COBYLA uses simulations to iteratively search for an optimized parameter set for a single waveform. Once trained, the neural density estimator in the SBI framework can be applied again on new time

series waveforms (that fall within the prior distribution) without retraining. As shown in the results, the posterior distribution is an object with several utilities. We emphasize the mapping between observations and ground truth parameters in this paper, but there are alternative uses such as parameter optimization via non-amortized inference [17], as well as building a more basic understanding of the model itself. Further, significant research efforts currently underway have the potential to decrease the computational cost of likelihood-free inference, making these techniques more accessible.

As an alternative to SBI, there has been substantial work in using probabilistic programming languages (PPL) to enable Bayesian inference on stochastic simulators. Models implemented in a PPL can be used in combination with modern inference algorithms (i.e. NUTS and ADVI) which have a dramatically lower computational cost compared to SBI [36, 62–64]. Unfortunately such approaches have the drawback that existing simulators must be completely rewritten in a PPL which may even compromise the efficiency of running forward simulations. Nevertheless, these techniques have proved highly useful for inference in large-scale neural simulators, and can potentially be combined with surrogate models that approximate the forward simulations of existing models.

## Other important future directions

In this study, we showed that PCA is the appropriate choice when compared to simple hand-crafted summary statistics when performing SBI on time series waveforms. However, PCA is constrained to preserve high-variance features, when in fact low-variance features may also be critical for identifying certain parameters. An important line of future work is the improvement of methods to learn summary statistics from neurophysiological signals that can help identify features of the signal that are essential for accurate parameter estimation. A promising development in this domain is the use of embedding networks that are trained to estimate summary statistics simultaneously with the neural density estimator used to approximate the posterior distribution that can account for those summary statistics [17, 22, 45, 46]. Currently, it is unclear if existing methods to train these embedding networks coupled to neural density estimators are sufficient and require further analysis. Our HNN-SBI examples focused on making inferences by constraining only to one output of the model; namely simulated current dipole waveforms. However, due to the multi-scale nature of the HNN model there are many other model outputs that could help constrain the inference problem, such as cell spiking profiles, and/or local field potential signals. A major advantage of the Bayesian framework is the ability to flexibly integrate multiple features into the parameter estimation. If additional multi-scale data is known, properties of this data can provide further summary statistics over which the inference problem can be constrained. An important limitation of the current study is a characterization of how observation noise impacts posterior parameter inferences, and which summary statistics are more or less robust to such noise. In our study, `PCA30` was found to perform best for parameter recovery, however the information gained from such low variance features may be highly sensitive to noise. Similarly, the hand-crafted summary statistics like `Peak` and `BandPower` may be particularly impacted by noise. The sensitivity of neural density estimators to observation noise, and more generally identifying model misspecification, is an open problem in the SBI field [16, 44] and an important direction for future research. Careful parameterization of observation noise, and establishment of predictive validity of the model using real data, will be critical next steps for robust inferences with the HNN-SBI framework.

## Conclusion

Using detailed neural models in a Bayesian framework is the product of significant developments in machine learning, biophysical modeling, and high-performance computing that have evolved largely independently. Our results demonstrate that large-scale biophysically detailed models, like HNN, are now amenable to Bayesian methods via the SBI framework, an approach that has not been feasible in the past. However, this novel combination produces new conceptual and technical challenges that must be addressed to effectively use these techniques. While in this work we provide guidelines for addressing such challenges, more research in the domains of neural modeling and likelihood-free inference are needed. It is apparent that the combination of HNN with SBI is a step forward for making mechanistic inferences underlying MEG/EEG biomarkers, with the potential to provide novel circuit-level predictions on disease and neural function. Our results lay the foundation for similar integration of SBI into the growing number of biophysically detailed neural modeling frameworks to advance neuroscience discovery.

## Acknowledgments

Simulations were implemented using HNN-core [65], a command line interface to the HNN model: https://github.com/jonescompneurolab/hnn-core.

Experiments were made possible thanks to the Python scientific ecosystem: Python [66], SBI [67], PyTorch [68], NumPy [69], SciPy [70], Matplotlib [71], Seaborn [72], and Dask [73].

We'd like to thank Ryan Thorpe for feedback on the manuscript, and Blake Caldwell for technical guidance early on in this study.

## Author Contributions

**Conceptualization:** Nicholas Tolley, Pedro L. C. Rodrigues, Alexandre Gramfort, Stephanie R. Jones.

**Data curation:** Nicholas Tolley.

**Formal analysis:** Nicholas Tolley, Pedro L. C. Rodrigues.

**Funding acquisition:** Alexandre Gramfort, Stephanie R. Jones.

**Investigation:** Nicholas Tolley.

**Methodology:** Nicholas Tolley.

**Software:** Nicholas Tolley, Pedro L. C. Rodrigues.

**Supervision:** Pedro L. C. Rodrigues, Alexandre Gramfort, Stephanie R. Jones.

**Visualization:** Nicholas Tolley.

**Writing – original draft:** Nicholas Tolley.

**Writing – review & editing:** Nicholas Tolley, Pedro L. C. Rodrigues, Alexandre Gramfort, Stephanie R. Jones.

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
