## [Decision Letter · Decision Letter 0]

24 Sep 2023

Dear Mr. Tolley,

Thank you very much for submitting your manuscript "Methods and considerations for estimating parameters in biophysically detailed neural models with simulation based inference" for consideration at PLOS Computational Biology.

As with all papers reviewed by the journal, your manuscript was reviewed by members of the editorial board and by several independent reviewers. In light of the reviews (below this email), we would like to invite the resubmission of a significantly-revised version that takes into account the reviewers' comments.

We cannot make any decision about publication until we have seen the revised manuscript and your response to the reviewers' comments. Your revised manuscript is also likely to be sent to reviewers for further evaluation.

Sincerely,

Lyle Graham

Section Editor

PLOS Computational Biology

Reviewer's Responses to Questions

**Comments to the Authors:**

Reviewer #1: This ms focuses on techniques for parameter inference of biophysically detailed models. This is challenging due to the high dimensionality of the parameter space. The ms attempts to address these challenges and offers an interesting overview of related work and literature. Below I am including some comments and suggestions for various parts of the ms.

Introduction

It is not clear to me how the “limited set of parameters …chosen”(l. 17) are chosen? If the current approach aims at overcoming the high number of parameters in detailed models (l.14), this is not achieved by picking a few parameters and fixing others as it is done in the Results, see also below.

DCM does not assume that parameters are independent as seemingly suggested on l.36 Interactions are described using the posterior correlation matrix, see e.g. K. E. Stephan, N. Weiskopf, P. M. Drysdale, P. A. Robinson, and K. J. Friston, “Comparing hemodynamic models with DCM.,” Neuroimage, vol. 38, no. 3, pp. 387–401, Nov. 2007. There is a mean field factorization over parameters and hyperparameters, not parameters, in the Variational Laplace inference scheme. Only one DCM variant, the so called mean field models (MFM), not others, assume that separate ensembles are coupled by mean field effects, see Marreiros, A. C., Kiebel, S. J., & Friston, K. J. (2010). A dynamic causal model study of neuronal population dynamics. Neuroimage, 51(1), 91-101. Similar remarks apply to comments in Discussion (see also below).

On a different matter, the lack of validation with empirical data is a significant limitation of the current ms. Even when using simulated data, like it is done here, one needs to consider noise in the observations (I could not find such reference in the ms). Even if this was used, it is not clear if it related to real noise. Various reasons could account for indeterminacy and lack of parameter identifiability of the sort discussed here, including low SNR or not enough data. This is normally taken care of by carefully parameterising observation noise (besides neural noise) and using real data to assess the predictive validity of models, i.e. whether inferences from one group predict responses of another. Also, different modalities, like LFP vs MEG will have different SNR and observation noise and the sensitivity of predicted and real data on biophysical parameters will depend on both parameters and observations. In brief, identifiability cannot only be assessed using simulations.

Methods

Could be expanded by moving material from elsewhere, see below.

Results

Again, it is not clear how this approach overcomes “challenges of…Bayesian inference” (l. 162) since only few parameters are estimated. Also, recovering original parameter estimates, known as face validity is discussed in many DCM papers. Thus, this criticism appears a bit misleading. Similarly, l. 295 and l. 574 should be toned down. DCM e.g. uses log normal priors, posterior correlations and a complexity term in the Free Energy alongside BMR (see also below) to describe interactions. It is not clear to me how indeterminacies of parameters are mitigated or addressed here. For example, in the last simulation considered, there is a clear overlap between drives to L2 neurons, EL2, IL”, quantified by relatively large OVL values 0.190 and 0.011 (l. 655) and is left unattended.

l.391-401 The discussion about the utility of summary statistics could be motivated better or have parts removed. Of course, the most informative feature in a linear system would be the maximum number of principal components (PCA30 here, nothing surprising). Similarly, since observations are ERPs, it is expected that band power will be less informative. Similarly, in the last example, only PCA30 results are considered (l. 640 and Figure 8 ) In general, what do we learn from the results about summary statistics besides that power is not an appropriate measure here? The authors should consider shortening the corresponding sections. Also when talking about PCA , variance explained should be added in all instances this is used.

l. 448 The simple linear RC circuit used as a first example has indeed some pedantic value. At the same time, it can be a bit misleading: the vast majority of interesting biophysical models are nonlinear, with much tighter posterior distributions, closer to the ground truth, like the one found here. In this case, interactions of the sort considered in the RC model are not of much practical relevance.

l. 613 Here and in other simulations only a few parameters are chosen: “parameters that define a small subset of local network …connectivity”. How is then this approach dealing with high dimensional parameter spaces as it seems to be the claim?

Discussion

l. 729 and below. Several parts need to be more accurate here. DCM does not employ a mean field assumption between parameters (see also remarks above regarding Introduction and Results) E-I balance has been discussed in several DCM papers, see e.g. Legon, W., Punzell, S., Dowlati, E., Adams, S. E., Stiles, A. B., & Moran, R. J. (2016). Altered prefrontal excitation/inhibition balance and prefrontal output: markers of aging in human memory networks. Cerebral Cortex, 26(11), 4315-4326. Also, Pinotsis, D. A., Perry, G., Litvak, V., Singh, K. D., & Friston, K. J. (2016). Intersubject variability and induced gamma in the visual cortex: DCM with empirical B ayes and neural fields. Human brain mapping, 37(12), 4597-4614.

Previous DCM work has used the HNN model, see Pinotsis, D. A., & Miller, E. K. (2020). Differences in visually induced MEG oscillations reflect differences in deep cortical layer activity. Communications biology, 3(1), 707, the authors present an approach for inferring parameters of detailed biophysical models based on statistical decision theory. It would be interesting to consider links between these approaches here. Another important development is the use of Bayesian Model Reduction that allows one to perform greedy search to compute model evidence in large spaces, see Friston, K., Parr, T., & Zeidman, P. (2018). Bayesian model reduction. arXiv preprint arXiv:1805.07092. This work should be discussed too.

Minor comments

l. 169-182, l. 235-247 These sections seem more appropriate for the Methods section.

l. 188 and below. It is a bit unclear what is step 1vs 2 in Steps 1-2 etc. Better break down each step separately.

l. 220 I guess here you refer to stochastic dynamics rendering ground truth parameters unknown. Better include a relevant discussion.

l.225 Is OVL more informative than other measures discussed (PRE,PPC)?

Fig3 and others. More detail is needed for Figure legends, e.g. there is no description for subpanels / e.g. various pair plots. Similarly for other Figure legends.

l. 401 what are the “desirable properties” implied here is unclear to me.

l. 488-495 (around these lines) and l. 597-600This could be more appropriate for the Discussion section.

l.564 -565 This seems to be an unnecessary repetition of the role of small distal variance in efficient inference.

l. 593 please add “of” after “strength”

l. 649 please use plural: “differences” not “difference”

l.787 the ending of the sentence sounds odd.

Reviewer #2: The paper introduces the use of simulation based inference for Biophysically detailed neural modeling, that was developed by authors to study the multi-scale neural origin of human MEG/EEG signals, namely the Human Neocortical Neurosolver (HNN). Although the HNN and SBI method have been previously introduced and the novelty of the work is modest, the paper provides an excellent guide to inference in HNN. SBI approach is flexible and efficient method for Bayesian estimation of model parameters, and it is a wise choice for such models used in this paper.

The paper is well-written with clear structure. In particular, it is the strong point of the paper to investigate the relation between parameters for non-identifiability issue and the existence of degeneracy in model, with the diagnostics for reliability of the inference. This will be very useful to the community.

Comments:

1) In abstract it would be clear to mention that the paper is focusing on simulated data, as a proof of concept. The SBI on empirical data will be more challenging.

Introduction:

2) Following DCM, it is worth mentioning the new generation of automatic inference using HMC and its alternatives (ADVI/ MAPs) in probabilistic programming languages (PPLs such as Stan, PyMc3) for system neuroscience, especially for inference in epilepsy. Of the challenges can be the efficient model's implementation in a specific language (in contrast to the flexibility of SBI), and differentiability due to the need for gradient calculation due to designed gradient-based algorithms in PPLs (NUTS/ADVI/MAP). In general, Bayesian inference using PPLs requires efficient reparameterization techniques as introduced by Jha et al, MLST 2022, doi: 10.1088/2632-2153/ac9037. Please salso ee:

Hashemi, Meysam, et al. "The Bayesian Virtual Epileptic Patient: A probabilistic framework designed to infer the spatial map of epileptogenicity in a personalized large-scale brain model of epilepsy spread." NeuroImage 217 (2020): 116839.

Hashemi, Meysam, et al. "On the influence of prior information evaluated by fully Bayesian criteria in a personalized whole-brain model of epilepsy spread." PLoS computational biology 17.7 (2021): e1009129.

Vattikonda, Anirudh N., et al. "Identifying spatio-temporal seizure propagation patterns in epilepsy using Bayesian inference." Communications biology 4.1 (2021): 1244.

Jha, Jayant, et al. "Fully Bayesian estimation of virtual brain parameters with self-tuning Hamiltonian Monte Carlo." Machine Learning: Science and Technology 3.3 (2022): 035016.

3) line 57:

Hashemi et al. 2023 has investigated a reduced representation of neural dynamics at each brain region (slow-fast system), but coupled at whole brain level (even by 400 connected regions). The challenges and advantages of SBI (difficulty in data features calculation, large number of simulations, but capability to address degeneracy and amortized strategy for rapid hypothesis evaluation at patient-level) have been discussed in details.

4) line 59: large-scale biophysical models;

Indeed, a situation that SBI is highly beneficial, is when the model is very high dimensional (eg., whole brain models) but a few number of parameters is required to be estimated, for instance in Higgs Boson discovery, Cramer PNAS, and the following refs in system neuroscience for prediction on healthy aging and Alzheimer's Disease, respectively:

Lavanga, Mario, et al. "The virtual aging brain: a model-driven explanation for cognitive decline in older subjects." bioRxiv (2022): 2022-02.

5) Section HNN (after line 101):

Since the paper is mainly designed to infer the parameters of HNN, it is easier for reader to some details on model eg in appendix, and what are the parameters (their dimension and their short description) as the target of estimation. I see Table 1, but it is worthy few lines to explain them in main text. What is the dimension of equations, (missing) variables, and parameters ti be estimated.

6) Section Posterior diagnostics

Is PRE able to accurately quantify the error in the case of multimodality?

Introducing PPC as the root mean squared error can introduces serious issues to quantify the quality of fit and the predictions, especially for noisy time series. I do not expect to changes the results, but at least it is expected to be point out in Discussion. The more principled way is to use rank plot, posterior shronkages, z-score (to incorporeal the uncertainty in prior), and WAIC (see Hashemi etl., PLOS CB 2021, https://doi.org/10.1371/journal.pcbi.1009129).

OVL is used to report the separability of posteriors across conditions? what is its advantage compared to KL or Ks distance?

7) Why the prior is transformed in the range of (0, 1)? Is this a sort of non-centered reparameterization? I did not get the motivation for transformation, and why for some linear/exponential.

8) Page 7, section The important role of summary statistics:

As the data features play a critical role, what are the exact features for MEG used in this study? In Fig 3, only PCA30 or with statistical moments? In the later former case, how the mean, std of observation can be inferred from a mean of I_input with only PCA? also peaks of time series of the power spectrum of time series? It is more clear to mention dimension of each feature. How is the fitted data for this example. How the peaks are calculated? For noisy signal this can be challenging to define a peak and calculate data features. PCA30 is robust for signal to noise ratio? A combination of all features will improve the inference compared to only PCA30?

9) The indeterminacies refers to the structural non-identifiability? (see https://doi.org/10.1016/j.neunet.2023.03.040 that distinguish structural and practical non-identifiability).

10) What is computational cost for the training and the dimension \\theta, x_o. The cost of simulations can be addressed by parallel running on HPC, which is a great advantage of SBI compared to MCMC methods.

Typos:

subscription in GABAb, line 115, and caption of Fig 2, line 616, etc

in is repeated in line 607.

**Have the authors made all data and (if applicable) computational code underlying the findings in their manuscript fully available?**

Reviewer #1: Yes

Reviewer #2: Yes

PLOS authors have the option to publish the peer review history of their article (what does this mean?). If published, this will include your full peer review and any attached files.

Reviewer #1: No

Reviewer #2: No
---

## [Decision Letter · Decision Letter 1]

10 Feb 2024

Dear Mr. Tolley,

We are pleased to inform you that your manuscript 'Methods and considerations for estimating parameters in biophysically detailed neural models with simulation based inference' has been provisionally accepted for publication in PLOS Computational Biology.

Best regards,

Lyle Graham

Section Editor

PLOS Computational Biology

Reviewer's Responses to Questions

**Comments to the Authors:**

Reviewer #1: The authors have addressed my comments in a satisfactory manner.

**Have the authors made all data and (if applicable) computational code underlying the findings in their manuscript fully available?**

Reviewer #1: Yes

PLOS authors have the option to publish the peer review history of their article (what does this mean?). If published, this will include your full peer review and any attached files.

Reviewer #1: **Yes: **Dimitris Pinotsis

---

## [Editor Report · Acceptance letter]

21 Feb 2024

PCOMPBIOL-D-23-00607R1 

Methods and considerations for estimating parameters in biophysically detailed neural models with simulation based inference

Dear Dr Tolley,

I am pleased to inform you that your manuscript has been formally accepted for publication in PLOS Computational Biology. Your manuscript is now with our production department and you will be notified of the publication date in due course.

With kind regards,

Judit Kozma
